# Hypercontractive Inequalities for the Second Norm of Highly Concentrated Functions, and Mrs. Gerber’s-Type Inequalities for the Second Rényi Entropy

**DOI:** 10.3390/e24101376

**Published:** 2022-09-27

**Authors:** Niv Levhari, Alex Samorodnitsky

**Affiliations:** 1School of Engineering and Computer Science, The Hebrew University of Jerusalem, Jerusalem 9103401, Israel; 2School of Mathematical Sciences, Tel Aviv University, Tel Aviv 6997801, Israel

**Keywords:** entropy, hypercontractivity, Rényi entropy, Mrs. Gerber’s inequality

## Abstract

Let $T_{\epsilon }$, 0≤ϵ≤1/2, be the noise operator acting on functions on the boolean cube ${\left\{0,1\right\}}^{n}$. Let *f* be a distribution on ${\left\{0,1\right\}}^{n}$ and let q>1. We prove tight Mrs. Gerber-type results for the second Rényi entropy of $T_{\epsilon }f$ which take into account the value of the $q^{th}$ Rényi entropy of *f*. For a general function *f* on ${\left\{0,1\right\}}^{n}$ we prove tight hypercontractive inequalities for the $\ell_{2}$ norm of $T_{\epsilon }f$ which take into account the ratio between $\ell_{q}$ and $\ell_{1}$ norms of *f*.

## 1. Introduction

This paper considers the problem of quantifying the decrease in the $\ell_{2}$ norm of a function on the boolean cube when this function is acted on by the noise operator.

Given a noise parameter 0≤ϵ≤1/2, the noise operator $T_{\epsilon }$ acts on functions on the boolean cube as follows: for $f:\;{\left\{0,1\right\}}^{n}\to R$, $T_{\epsilon }f$ at a point *x* is the expected value of *f* at *y*, where *y* is a random binary vector whose $i^{th}$ coordinate is $x_{i}$ with probability 1−ϵ and $1-x_{i}$ with probability ϵ, independently for different coordinates. Namely, $\left(T_{\epsilon }f\right)\left(x\right)=\sum_{y\in {\left\{0,1\right\}}^{n}}\epsilon^{\left|y-x\right|}{\left(1-\epsilon \right)}^{n-|y-x|}f\left(y\right)$, where |·| denotes the Hamming distance. We will write $f_{\epsilon }$ for $T_{\epsilon }f$, for brevity.

Note that $f_{\epsilon }$ is a convex combination of shifted copies of *f*. Hence, the noise operator decreases norms. Recall that the $\ell_{q}$ norm of a function is given by ${∥f∥}_{q}={\left({E|f|}^{q}\right)}^{\frac{1}{q}}$ (the expectations here and below are taken w.r.t. the uniform measure on ${\left\{0,1\right\}}^{n}$). The norms ${\{∥f∥}_{q}{\}}_{q}$ increase with *q*. An effective way to quantify the decrease of $\ell_{q}$ norm under noise is given by the hypercontractive inequality [1,2,3] (see also, e.g., [4] for background), which upperbounds the $\ell_{q}$ norm of the noisy version of a function by a smaller norm of the original function.
(1)$$∥f_{\epsilon }{∥}_{q}\;\leq \;{∥f∥}_{1+{\left(1-2\epsilon \right)}^{2}\left(q-1\right)}.$$
This inequality is essentially tight in the following sense. For any $p<1+\left(q-1\right){\left(1-2\epsilon \right)}^{2}$ there exists a non-constant function $f:{\left\{0,1\right\}}^{n}\to R$ with $∥f_{\epsilon }{∥}_{q}>{∥f∥}_{p}$.

*Entropy* provides another example of a convex homogeneous functional on (nonnegative) functions on the boolean cube. For a nonnegative function *f* let the entropy of *f* be given by $Ent\left(f\right)=Ef\log_{2}f-Ef\log_{2}Ef$. The entropy of *f* is closely related to Shannon’s entropy of the corresponding distribution f/Σf on ${\left\{0,1\right\}}^{n}$, and similarly the entropy of $f_{\epsilon }$ is related to Shannon’s entropy of the output of a binary symmetric channel with error probability ϵ on input distributed according to f/Σf (see below and, e.g., the discussion in the introduction of [5]). The decrease in entropy (or, correspondingly, the increase in Shannon’s entropy) after noise is quantified in the “Mrs. Gerber’s Lemma” [6]:(2)$$Ent\left(f_{\epsilon }\right)\;\leq \;nEf\cdot \psi \left(\frac{Ent(f)}{nEf},\;\epsilon \right),$$
where $\psi =\psi \left(x,\epsilon \right)=H\left(\left(1-2\epsilon \right)\cdot H^{-1}\left(1-x\right)\;+\;\epsilon \right)$ is an explicitly given function on $\left[0,1\right]\times \left[0,1/2\right]$, which is increasing and strictly concave in its first argument for any $0<\epsilon <\frac{1}{2}$. Here and below we write $H\left(t\right)=t\log_{2}\left(\frac{1}{t}\right)+\left(1-t\right)\log_{2}\left(\frac{1}{1-t}\right)$ for the binary entropy function.

Equality holds iff *f* is a product function with equal marginals. That is, there exists a function g:{0,1}→R, such that for any $x=\left(x_{1},\ldots ,x_{n}\right)\in {\left\{0,1\right\}}^{n}$ holds $f\left(x\right)=\prod_{i=1}^{n}g\left(x_{i}\right)$.

One has ψ(0,ϵ)=0 and ${\frac{\partial \psi }{\partial x}}_{|x=0}={\left(1-2\epsilon \right)}^{2}$. Hence $\psi \left(x,\epsilon \right)\leq {\left(1-2\epsilon \right)}^{2}\cdot x$, with equality only at x=0. Hence the inequality (Equation 2) has the following weaker linear approximation version
(3)$$Ent\left(f_{\epsilon }\right)\;\leq \;{\left(1-2\epsilon \right)}^{2}\cdot Ent\left(f\right),$$
in which equality holds if and only if *f* is a constant function.

*Rényi entropies*. There is a well-known connection between $\ell_{q}$ norms of a nonnegative function *f* and its entropy (see, e.g., [7]): Assume, as we may by homogeneity, that Ef=1. Then $Ent\left(f\right)=\lim_{q\to 1}\frac{1}{q-1}\log_{2}{\left||f|\right|}_{q}^{q}$. (The quantity $Ent_{q}\left(f\right)=\frac{1}{q-1}\log_{2}{\left||f|\right|}_{q}^{q}$ is known as the $q^{th}$ Rényi entropy of *f* ([8])). (Note that this notion is defined for all, not necessarily nonnegative, functions on ${\left\{0,1\right\}}^{n}$.) The entropies ${\left\{Ent_{q}\left(f\right)\right\}}_{q}$ increase with *q*. Restating the inequality (Equation 1) in terms of Rényi entropies gives
$$Ent_{q}\left(f_{\epsilon }\right)\leq \frac{{\left(1-2\epsilon \right)}^{2}q}{{\left(1-2\epsilon \right)}^{2}\left(q-1\right)+1}\cdot Ent_{1+{\left(1-2\epsilon \right)}^{2}\left(q-1\right)}\left(f\right).$$

Note that taking q→1 in this inequality recovers only the (weaker) linear approximation version (Equation 3) of Mrs. Gerber’s inequality (Equation 2). This highlights an important difference between inequalities (Equation 1) and (Equation 2). Mrs. Gerber’s lemma takes into account the distribution of a function, specifically the ratio between its entropy and its $\ell_{1}$ norm. When this ratio is exponentially large in *n*, which typically holds in the information theory contexts in which this inequality is applied, (Equation 2) is significantly stronger than (Equation 3). On the other hand, hypercontractive inequalities seem to be typically applied in contexts in which the ratio between different norms of the function is subexponential in *n*, and there are examples of such functions for which (Equation 1) is essentially tight. With that, there are several recent results [9,10,11] which show that (Equation 1) can be strengthened, if the ratio $\frac{{∥f∥}_{q}}{{∥f∥}_{1}}$, for some q>1, is exponentially large in *n*. In the framework of Rényi entropies, the possibility of a result analogous to (Equation 2) for higher Rényi entropies was discussed in [12].

*Our results*. This paper proves a Mrs. Gerber type result for the second Rényi entropy, and a hypercontractive inequality for the $\ell_{2}$ norm of $f_{\epsilon }$ which take into account the ratio between $\ell_{q}$ and $\ell_{1}$ norms of *f*. We try to pattern the results below after (Equation 2).

We start with a Mrs. Gerber type inequality.

**Proposition** **1.**
*Let q>1, and let f be a nonnegative function on ${\left\{0,1\right\}}^{n}$ such that Ef=1. Then*

(4)
$$\frac{Ent_{2}\left(f_{\epsilon }\right)}{n}\;\leq \;\psi_{2,q}\left(\frac{Ent_{q}\left(f\right)}{n},\;\epsilon \right),$$

*where $\psi_{2,q}$ is an explicitly given function on $\left[0,1\right]\times \left[0,1/2\right]$, which is increasing and concave in its first argument. The function $\psi_{2,q}$ is defined in Definition 1 below.*

*This inequality is essentially tight in the following sense. For any 0<x<1 and $0<\epsilon <\frac{1}{2}$, and for any $y<\psi_{2,q}\left(x,\epsilon \right)$ there exists a sufficiently large n and a nonnegative function f on ${\left\{0,1\right\}}^{n}$ with Ef=1, $\frac{Ent_{q}\left(f\right)}{n}\leq x$ and $\frac{Ent_{2}\left(f_{\epsilon }\right)}{n}>y$.*


Let us make some comments about this result.

–The functions ${\left\{\psi_{2,q}\right\}}_{q}$ are somewhat cumbersome to describe, and hence we relegate their precise definition to Definition 1 below.–Inequality (Equation 4) upper bounds $Ent_{2}\left(f_{\epsilon }\right)$ in terms of $Ent_{q}\left(f\right)$ for q>1, and ϵ. Taking q=2 gives an upper bound on $Ent_{2}\left(f_{\epsilon }\right)$ in terms of $Ent_{2}\left(f\right)$ and ϵ, in analogy to (Equation 2).–Recall that for a point $x\in {\left\{0,1\right\}}^{n}$ and 0≤r≤n, the Hamming sphere of radius *r* around *x* is the set $\left\{y\in {\left\{0,1\right\}}^{n}:\;|y-x|=r\right\}$. As will be seen from the proof of Proposition 1, (Equation 4) is essentially tight for a certain convex combination of the uniform distribution on ${\left\{0,1\right\}}^{n}$ and the characteristic function of a Hamming sphere of an appropriate radius (depending on *q*, ϵ, and the required value of $Ent_{q}\left(f\right)$).–In information theory one typically considers a slightly different notion of Rényi entropies: For a probability distribution *P* on Ω, the $q^{th}$ Renyi entropy of *P* is given by $H_{q}\left(P\right)=-\frac{1}{q-1}\log_{2}\left(\sum_{\omega \in \Omega }P^{q}\left(\omega \right)\right)$. To connect notions, if *f* is a nonnegative (non-zero) function on ${\left\{0,1\right\}}^{n}$ with expectation 1, then $P=\frac{f}{2^{n}}$ is a probability distribution, and $Ent_{q}\left(f\right)=n-H_{q}\left(P\right)$. Furthermore, $Ent_{q}\left(f_{\epsilon }\right)=n-H_{q}\left(X⊕Z\right)$, where *X* is a random variable on ${\left\{0,1\right\}}^{n}$ distributed accordinng to *P* and *Z* is an independent noise vector corresponding to a binary symmetric channel with crossover probability ϵ. Hence, (Equation 2) can be restated as
$$H\left(X⊕Z\right)\geq n\cdot \varphi \left(\frac{H(X)}{n},\;\epsilon \right),$$
and Proposition 1 can be restated as
$$H_{2}\left(X⊕Z\right)\geq n\cdot \varphi_{2,q}\left(\frac{H_{q}\left(X\right)}{n},\;\epsilon \right)$$Here φ is an explicitly given function on $\left[0,1\right]\times \left[0,1/2\right]$, which is increasing and convex in its first argument (φ(x,ϵ)=1−ψ(1−x,ϵ)), and similarly for $\varphi_{2,q}$.Next, we describe our main result, a hypercontractive inequality for the $\ell_{2}$ norm of $f_{\epsilon }$ which takes into account the ratio between $\ell_{q}$ and $\ell_{1}$ norms of *f*, and more specifically $Ent_{q}\left(\frac{f}{{∥f∥}_{1}}\right)=\frac{q}{q-1}\log_{2}\left(\frac{{∥f∥}_{q}}{{∥f∥}_{1}}\right)$.

**Theorem** **1.**
*Let q>1, and let f be a non-zero function on ${\left\{0,1\right\}}^{n}$. Then*

(5)
$$∥f_{\epsilon }{∥}_{2}\;\leq \;{∥f∥}_{\kappa },$$

*where $\kappa =\kappa_{2,q}\left(\frac{Ent_{q}\left(\frac{f}{{∥f∥}_{1}}\right)}{n},\epsilon \right)$, and $\kappa_{2,q}$ is an explicitly given function on $\left[0,1\right]\times \left[0,1/2\right]$, which is decreasing in its first argument and which satisfies $\kappa_{2,q}\left(0,\epsilon \right)=1+{\left(1-2\epsilon \right)}^{2}$, for all $0\leq \epsilon \leq \frac{1}{2}$. The function $\kappa_{2,q}$ is defined in Definition 1 below.*

*This inequality is essentially tight in the following sense. For any 0<x<1 and $0<\epsilon <\frac{1}{2}$, and for any $y<\kappa_{2,q}\left(x,\epsilon \right)$ there exists a sufficiently large n and a function f on ${\left\{0,1\right\}}^{n}$ with $\frac{Ent_{q}\left(f/{∥f∥}_{1}\right)}{n}\geq x$ and $∥f_{\epsilon }{∥}_{2}>{∥f∥}_{y}$.*


Some comments (see also Lemma 10 below).

–The precise definition of the functions ${\left\{\kappa_{2,q}\right\}}_{q}$ will be given in Definition 1 below. At this point let us just observe that since the sequence ${\left\{Ent_{q}\left(f\right)\right\}}_{q}$ increases with *q*, we would expect the fact that $Ent_{q}\left(f\right)$ is large to become less significant as *q* increases. This is expressed in the properties of the functions ${\left\{\kappa_{2,q}\right\}}_{q}$ in the following manner: If q≥2 then for any $0<\epsilon <\frac{1}{2}$ the function $\kappa_{2,q}\left(x,\epsilon \right)$ starts as a constant-$\left(1+{\left(1-2\epsilon \right)}^{2}\right)$ function up to some x=x(q,ϵ)>0, and becomes strictly decreasing after that. In other words x(q,ϵ) is the largest possible value of $\frac{Ent_{q}\left(\frac{f}{{∥f∥}_{1}}\right)}{n}$ for which Theorem 1 provides no new information compared to (Equation 1). For 1<q<2 there is a value $0<\epsilon \left(q\right)<\frac{1}{2}$, such that for all ϵ≤ϵ(q) the function $\kappa_{2,q}\left(x,\epsilon \right)$ is strictly decreasing (in which case we say that x(q,ϵ)=0). However, x(q,ϵ)>0 for all ϵ>ϵ(q). The function ϵ(q) decreases with *q* (in particular, ϵ(q)=0 for g≥2). The function x(q,ϵ) increases both in *q* and in ϵ.–Notably, taking q→1 in Theorem 1 gives (see Corollary 1)
$$∥f_{\epsilon }{∥}_{2}\;\leq \;{∥f∥}_{\kappa },$$
where $\kappa =\kappa_{2,1}\left(Ent\left(\frac{f}{{∥f∥}_{1}}\right)/n,\epsilon \right)=-\frac{Ent\left(\frac{f}{{∥f∥}_{1}}\right)/n}{\phi_{\epsilon }\left(1-Ent\left(\frac{f}{{∥f∥}_{1}}\right)/n\right)}$. The function $\kappa_{2,1}\left(x,\epsilon \right)=-\frac{x}{\phi_{\epsilon }\left(1-x\right)}$ is strictly decreasing in *x* for any $0<\epsilon <\frac{1}{2}$. It satisfies $\kappa_{2,1}\left(0,\epsilon \right)=\lim_{x\to 0}\kappa_{2,1}\left(x,\epsilon \right)=1+{\left(1-2\epsilon \right)}^{2}$, for all $0\leq \epsilon \leq \frac{1}{2}$. Hence, this is stronger than (Equation 1) for any non-constant function *f* and for any $0<\epsilon <\frac{1}{2}$, with the difference between the two inequalities becoming significant when $Ent\left(\frac{f}{{∥f∥}_{1}}\right)/n$ is bounded away from 0.–As will be seen from the proof of Theorem 1, (Equation 5) is essentially tight for a certain convex combination of the uniform distribution on ${\left\{0,1\right\}}^{n}$ and characteristic functions of one or two Hamming spheres of appropriate radii (the number of the spheres and their radii depend on *q*, ϵ, and the required value of $Ent_{q}\left(\frac{f}{{∥f∥}_{1}}\right)$).–Let *f* be a non-constant function and let $0<\epsilon <\frac{1}{2}$ be fixed. Consider the function $F\left(q\right)=F_{f,\epsilon }\left(q\right)=\kappa_{2,q}\left(\frac{Ent_{q}\left(\frac{f}{{∥f∥}_{1}}\right)}{n},\epsilon \right)$. It will be seen that there is a unique value $1<q\left(f,\epsilon \right)\leq 1+{\left(1-2\epsilon \right)}^{2}$ of *q* for which F(q)=q. Furthermore, $q\left(f,\epsilon \right)=\min_{q\geq 1}F\left(q\right)$. Hence it provides the best possible value for κ in Theorem 1. With that, determining q(f,ϵ) might in principle require knowledge of all the Renyi entropies $Ent_{q}\left(f\right)$, for $1\leq q\leq 1+{\left(1-2\epsilon \right)}^{2}$, while typically we are in possession of one of the “easier” Rényi entropies, such as Ent(f) or $Ent_{2}\left(f\right)$.

### 1.1. Full Statements of Proposition 1 and Theorem 1

We now define the functions ${\left\{\psi_{2,q}\right\}}_{q}$ in Proposition 1 and ${\left\{\kappa_{2,q}\right\}}_{q}$ in Theorem 1, completing the statements of these claims. We start with introducing yet another function on $\left[0,1\right]\times \left[0,1/2\right]$ which will play a key role in what follows (we remark that this function was studied in [9]). For 0≤x≤1 and $0\leq \epsilon \leq \frac{1}{2}$, let $\sigma =H^{-1}\left(x\right)$ and let $y=y\left(x,\epsilon \right)=\frac{-\epsilon^{2}+\epsilon \sqrt{\epsilon^{2}+4\left(1-2\epsilon \right)\sigma \left(1-\sigma \right)}}{2(1-2\epsilon )}$. Let
$$\Phi \left(x,\epsilon \right)\;=\;\frac{1}{2}\cdot \left(x-1+\sigma H\left(\frac{y}{\sigma }\right)+\left(1-\sigma \right)H\left(\frac{y}{1-\sigma }\right)+2y\log_{2}\left(\epsilon \right)+\left(1-2y\right)\log_{2}\left(1-\epsilon \right)\right).$$

The function Φ is nonpositive. It is increasing and concave in its first argument. Additional relevant properties of Φ are listed in Lemma 3 below. For a fixed ϵ, it will be convenient to write $\phi_{\epsilon }\left(x\right)=\Phi \left(x,2\epsilon (1-\epsilon )\right)$, viewing $\phi_{\epsilon }$ as a univariate function on [0,1].

**Definition** **1.**
*Let 0≤x≤1 and $0\leq \epsilon \leq \frac{1}{2}$.*


*If $\phi_{\epsilon }^{\prime }\left(1-x\right)<\frac{1}{q}$, let $\alpha_{0}={\left(\phi_{\epsilon }^{\prime }\right)}^{-1}\left(\frac{1}{q}\right)$. Define*

$$\psi_{2,q}\left(x,\epsilon \right)\;=\;2\cdot \left\{\begin{array}{ccc} \frac{q-1}{q}\cdot x+\left(\phi_{\epsilon }\left(\alpha_{0}\right)+\frac{1-\alpha_{0}}{q}\right) & if & \phi_{\epsilon }^{\prime }\left(1-x\right)<\frac{1}{q}\\ \phi_{\epsilon }\left(1-x\right)+x & otherwise \end{array}\right.$$


*Let $y=\frac{q-1}{q}\cdot x+\frac{1}{q}$. Let $q_{0}=1+{\left(1-2\epsilon \right)}^{2}$. If $y\geq \frac{1}{q_{0}}$, let $\alpha_{0}$ be determined by $1-\alpha_{0}-\frac{\alpha_{0}\phi_{\epsilon }\left(\alpha_{0}\right)}{1-\alpha_{0}}=y$. If x=0, define $\kappa_{2,q}\left(x,\epsilon \right)=q_{0}$. Otherwise, define*

$$\kappa_{2,q}\left(x,\epsilon \right)\;=\;\left\{\begin{array}{ccc} q_{0} & if & y\leq \frac{1}{q_{0}}\\ -\frac{x}{\phi_{\epsilon }\left(1-x\right)} & if & y>\frac{1}{q_{0}}\;and\;-\frac{x}{\phi_{\epsilon }\left(1-x\right)}\geq \;q\\ \frac{\alpha_{0}-1}{\phi_{\epsilon }\left(\alpha_{0}\right)} & if & y>\frac{1}{q_{0}}\;and\;-\frac{x}{\phi_{\epsilon }\left(1-x\right)}<q \end{array}\right.$$




We remark that it is not immediately obvious that the functions $\psi_{2,q}$ and $\kappa_{2,q}$ are well-defined. This will be clarified in the proofs of Proposition 1 and Theorem 1.

We state explicitly some special cases of Theorem 1, which seem to be the most relevant for applications. They describe the improvement over (Equation 1), given non-trivial information about Ent(f) and ${∥f∥}_{2}$.

**Corollary** **1.**
*1.* 
*Taking q→1 in Theorem 1 gives:*

$$∥f_{\epsilon }{∥}_{2}\;\leq \;{∥f∥}_{\kappa },\;with\;\kappa =-\frac{Ent\left(\frac{f}{{∥f∥}_{1}}\right)/n}{\phi_{\epsilon }\left(1-Ent\left(\frac{f}{{∥f∥}_{1}}\right)/n\right)}.$$

*2.* 
*Taking q=2 in Theorem 1 gives, for $x=\frac{Ent_{2}\left(\frac{f}{{∥f∥}_{1}}\right)}{n}$ and $q_{0}=1+{\left(1-2\epsilon \right)}^{2}$*

$$∥f_{\epsilon }{∥}_{2}\;\leq \;{∥f∥}_{\kappa },\;with\;\kappa =\left\{\begin{array}{ccc} q_{0} & if & \frac{x+1}{2}\leq \frac{1}{q_{0}}\\ \frac{\alpha -1}{\phi_{\epsilon }\left(\alpha \right)} & otherwise \end{array}\right.$$


*In the second case α is determined by $1-\alpha -\frac{\alpha \phi_{\epsilon }\left(\alpha \right)}{1-\alpha }=\frac{x+1}{2}$.*



We observe that both Proposition 1 and Theorem 1 are based on the following claim ([9], Corollary 3.2). This claim also explains the relevance of function Φ.

**Theorem** **2.**
*Let 0≤x≤1. Let f be a function on ${\left\{0,1\right\}}^{n}$ supported on a set of cardinality at most $2^{xn}$. Then, for any $0\leq \epsilon \leq \frac{1}{2}$ holds*

$$\langle f_{\epsilon },f\rangle \;\leq \;2^{\left(2\Phi \left(x,\epsilon \right)+1-x\right)\cdot n}\cdot {∥f∥}_{2}^{2},$$

*Moreover, this is tight, up to a polynomial in n factor, if f is the characteristic function of a Hamming sphere of radius $H^{-1}\left(x\right)\cdot n$.*


### 1.2. Applications

We describe some applications of the results above, related mainly to coding theory. We start with providing some relevant context.

*Coding theory.* A binary error-correcting code *C* of length *n* and minimal distance *d* is a subset of ${\left\{0,1\right\}}^{n}$ in which the distance between any two distinct points is at least *d*. Let A(n,d) be the maximal size of such a code. A well-known open problem in coding theory is to determine, given $0<\delta <\frac{1}{2}$, the *asymptotic maximal rate*
$R\left(\delta \right)={\mathrm{lim sup}}_{n\to \infty }\frac{1}{n}\log_{2}A\left(n,\lfloor \delta n\rfloor \right)$ of a code with relative distance δ. The best known lower bound on R(δ) is the Gilbert-Varshamov bound R(δ)≥1−H(δ) [13]. The best known upper bounds on R(δ) were obtained in [14] using the linear programming relaxation, constructed in [15], of the combinatorial problem of bounding A(n,d). Let $A_{LP}\left(n,d\right)$ be the value of the appropriate linear program of [15] and let $R_{LP}\left(\delta \right)={\mathrm{lim sup}}_{n\to \infty }\frac{1}{n}\log_{2}A_{LP}\left(n,\lfloor \delta n\rfloor \right)$. By construction, $A_{LP}\left(n,d\right)\geq A\left(n,d\right)$ for all *n* and *d* and hence $R_{LP}\left(\delta \right)\geq R\left(\delta \right)$. The *first JPL bound* of [14] is $R\left(\delta \right)\leq R_{LP}\left(\delta \right)\leq H\left(1/2-\sqrt{\delta (1-\delta )}\right)$. This bound is the best known for a subrange of values of δ. The best known bound is the *second JPL bound* of [14]. It is better than the first bound for relatively small values of δ. However, it is more complicated to state explicitly and we omit it here. The second JPL bound is strictly larger than the Gilbert-Varshamov bound for all $0<\delta <\frac{1}{2}$, and hence R(δ) is unknown for all these values of δ.

The value of $R_{LP}\left(\delta \right)$ is also unknown, for all $0<\delta <\frac{1}{2}$. Clearly $R_{LP}\left(\delta \right)\geq R\left(\delta \right)\geq 1-H\left(\delta \right)$. It was conjectured in [14] that $R_{LP}\left(\delta \right)$ lies strictly between the second JPL bound and the Gilbert-Varshamov bound. On the other hand, there is a convincing numeric evidence [16] that $R_{LP}\left(\delta \right)$ in fact coincides with the second JPL bound. A lower bound $R_{LP}\left(\delta \right)\geq \frac{1-H\left(\delta \right)+H\left(1/2-\sqrt{\delta (1-\delta )}\right)}{2}$ was shown in [17] (note that the RHS here is the arithmetic average of the Gilbert-Varshamov bound and the first JPL bound). It was improved, for a subrange of δ, in [18].

A different approach to obtain upper bounds on the cardinality of binary codes was presented in [19]. For a subset $D\subseteq {\left\{0,1\right\}}^{n}$, let $M_{D}$ be the adjacency matrix of the subgraph of the discrete cube induced by the vertices of *D*. Let λ(D) be the maximal eigenvalue of $M_{D}$. The following claim was proved in [19] for binary linear codes (and extended in [18] to general binary codes): Let *D* be subset of ${\left\{0,1\right\}}^{n}$ with λ(D)≥n−2d+1. Let *C* be a code of length *n* and minimal distance *d*. Then |C|≲|D| (here we use the approximate inequality sign to indicate that the inequality holds up to lower order terms). Choosing for *D* the Hamming balls of different radii with their corresponding parameters leads to a simple proof of the first JPL bound on R(δ). Ref. [19] posed the natural problem of finding subsets of ${\left\{0,1\right\}}^{n}$ with the largest possible eigenvalue for their cardinality. This question was answered in [20], where is was shown that Hamming balls of radius r=ρn, $0<\rho <\frac{1}{2}$ have essentially the largest eigenvalues for their cardinality. This seems to indicate that at least the straightforward version of the approach of [19], as described above, does not lead to an improvement of the first JPL bound. The claim in [20] was derived from a *logarithmic Sobolev inequality* for highly concentrated functions on the boolean cube. We continue with a brief description of relevant notions.

*Logarithmic Sobolev inequalities*. Viewing both sides of (Equation 1) as functions of ϵ, and writing L(ϵ) for the LHS and R(ϵ) for the RHS, we have $L\left(0\right)=R\left(0\right)={∥f∥}_{2}$, and L(ϵ)≤R(ϵ) for $0\leq \epsilon \leq \frac{1}{2}$. Since both *L* and *R* are differentiable in ϵ this implies $L^{\prime }\left(0\right)\leq R^{\prime }\left(0\right)$. This inequality is the logarithmic Sobolev inequality ([3]) for the Hamming cube. We proceed to describe it in more detail. Recall that the Dirichlet form E(f,g) for functions *f* and *g* on the Hamming cube is defined by $E\left(f,g\right)=E_{x}\sum_{y∼x}\left(f(x)-f(y)\right)\left(g(x)-g(y)\right)$. Here y∼x means that *x* and *y* differ in precisely one coordinate. The logarithmic Sobolev inequality then states that $E\left(f,f\right)\geq 2\ln2\cdot Ent\left(f^{2}\right)$. This inequality describes the behavior of the norm on the RHS of the hypercontractive inequality (Equation 1) as ϵ→0 and, as such, can be viewed as a special case of (Equation 1). In point of fact, it was introduced in [3] as a way to prove (Equation 1) by (roughly speaking) integrating this inequality over the noise parameter (using the semigroup property of noise operators). Following [3], logarithmic Sobolev inequalities were shown to hold in many spaces of interest (see [21] for discussion and for many applications of these inequalities).

The logarithmic Sobolev inequality for highly concentrated functions in [20] (we will state this inequality explicitly in the discussion following Corollary 2 below) improves over the inequality $E\left(f,f\right)\geq 2\ln2\cdot Ent\left(f^{2}\right)$ similarly to the improvement to (Equation 1) provided by Theorem 1. However, deducing a tight hypercontractive inequality, such as Theorem 1, from the inequality in [20] by integration (following the approach of [3]) seems to be more challenging. Roughly speaking, the problem lies in the fact that the concentration of *f* might decrease very quickly under noise. With that, a family of logarithmic Sobolev inequalities, generalizing that of [20] was proved in [10]. Integrating these inequalities over noise leads to a family of hypercontractive inequalities which improve over (Equation 1) for highly concentrated functions and which are essentially tight in the vicinity of ϵ=0. These inequalities were used in [10] to prove a version of the *uncertainty principle* on ${\left\{0,1\right\}}^{n}$.

*An uncertainty principle on ${\left\{0,1\right\}}^{n}$*. We recall some basic notions in Fourier analysis on the Hamming cube (see [4]). For $\alpha \in {\left\{0,1\right\}}^{n}$, define the Walsh-Fourier character $W_{\alpha }$ on ${\left\{0,1\right\}}^{n}$ by setting $W_{\alpha }\left(y\right)={\left(-1\right)}^{\sum \alpha_{i}y_{i}}$, for all $y\in {\left\{0,1\right\}}^{n}$. The *weight* of the character $W_{\alpha }$ is the Hamming weight |α| of α. The characters ${\left\{W_{\alpha }\right\}}_{\alpha \in {\left\{0,1\right\}}^{n}}$ form an orthonormal basis in the space of real-valued functions on ${\left\{0,1\right\}}^{n}$, under the inner product $\langle f,g\rangle =\frac{1}{2^{n}}\sum_{x\in {\left\{0,1\right\}}^{n}}f\left(x\right)g\left(x\right)$. The expansion $f=\sum_{\alpha \in {\left\{0,1\right\}}^{n}}\hat{f}\left(\alpha \right)W_{\alpha }$ defines the Fourier transform $\hat{f}$ of *f*. We also have the Parseval identity, ${∥f∥}_{2}^{2}=\sum_{\alpha \in {\left\{0,1\right\}}^{n}}{\hat{f}}^{2}\left(\alpha \right)$.

Uncertainty principle asserts that a function and its Fourier transform cannot be simultaneously narrowly concentrated. A well-known way (see, e.g., [22]) to state this for the Hamming cube is as follows. If *f* is a non-zero function on ${\left\{0,1\right\}}^{n}$ then $\left|supp\left(f\right)\right|\geq \frac{2^{n}}{|supp\left(\hat{f}\right)|}$. In [10], (see also the discussion following Theorem 1.10 in [9]) a different way to formalize this statement for the Hamming cube was presented. If *f* is a function on ${\left\{0,1\right\}}^{n}$ with $\frac{Ent_{2}\left(\frac{f}{{∥f∥}_{1}}\right)}{n}\geq 1-H\left(\rho \right)$, then its Fourier transform $\hat{f}$ cannot attain its $\ell_{2}$ norm in a Hamming ball of radius much smaller than $\left(\frac{1}{2}-\sqrt{\rho (1-\rho )}\right)\cdot n$. This result was then used to establish some properties of binary linear codes.

*Our results*. We now pass to presenting our results which are relevant to the topics above. We first remark that the idea of using hypercontractivity to study binary codes was discussed already in [23]. In [24], the hypercontractive inequality (Equation 1) was used to obtain bounds on the distance components and other parameters of binary codes. We observe (a similar observation was made in [9]) that these bounds can be strengthened by replacing (Equation 1) by (stronger) inequalities of Theorem 1. We do not go into details.

Next, we consider some implications of Theorem 1, focussing on the behavior of the norm $\kappa =\kappa_{2,2}$ for values of the noise parameter ϵ in the vicinity of 0. Clearly, for any 0≤x≤1 the function $\kappa_{2,2}\left(x,\epsilon \right)$ is 2 at ϵ=0. We prove the following technical claim.

**Lemma** **1.**
*Assume 0<x<1. Let $\kappa \left(\epsilon \right)=\kappa_{2,2}\left(x,\epsilon \right)$.*

*1.* 

$$\kappa^{\prime }\left(0\right)\;=\;\frac{4}{\ln2}\cdot \frac{\left(2\sqrt{H^{-1}\left(1-x\right)\left(1-H^{-1}\left(1-x\right)\right)}-1\right)}{x}.$$

*2.* 
*Let ϵ∼0 express the fact that ϵ is a sufficiently small absolute constant. Then for ϵ∼0 holds $|\kappa^{\prime }\left(\epsilon \right)-\kappa^{\prime }\left(0\right)|\leq O\left(\epsilon \right)$, where the asymptotic notation hides absolute constants which may depend on x.*



We use the first part of this claim to rederive a slightly weaker (but sufficient for applications, see the dicussion following Corollary 4) version of the logarithmic Sobolev inequality from [20].

**Corollary** **2.**
*For any function f on ${\left\{0,1\right\}}^{n}$ holds*

$$E\left(f,f\right)\;\geq \;\ell \left(\frac{Ent_{2}\left(\frac{f}{{∥f∥}_{1}}\right)}{n}\right)\cdot Ent\left(f^{2}\right),$$

*where $\ell \left(x\right)=2\cdot \frac{1-2\sqrt{H^{-1}\left(1-x\right)\left(1-H^{-1}\left(1-x\right)\right)}}{x}$ is a convex and increasing function on [0,1], taking [0,1] onto $\left[2\ln2,2\right]$.*


We remark that in [20] (see also Theorem 6 in [10]) a somewhat stronger logarithmic Sobolev inequality $E\left(f,f\right)\geq \ell \left(\frac{Ent\left(\frac{f^{2}}{{∥f∥}_{2}^{2}}\right)}{n}\right)\cdot Ent\left(f^{2}\right)$ was shown using a different approach. (It seems that it might be possible to recover this stronger inequality by differentiating a corresponding hypercontractive inequality at zero, if one considers a more general version of Theorem 1 which takes into account the ratio between $\ell_{q}$ and $\ell_{p}$ norms of *f*, for q>p, and in this case taking both *q* and *p* to be very close to 2. We omit the details.)

Next, we use the second part of Lemma 1 to rederive the uncertainty principle from [10].

**Corollary** **3.**
*Let f be a non-zero function on ${\left\{0,1\right\}}^{n}$ such that $\frac{Ent_{2}\left(\frac{f}{{∥f∥}_{1}}\right)}{n}=1-H\left(\rho \right)$, for some 0≤ρ<1. Let $0\leq \mu <\frac{1}{2}-\sqrt{\rho (1-\rho )}$. Then*

$$\sum_{|\alpha |\leq \mu n}{\hat{f}}^{2}\left(\alpha \right)\;\leq \;2^{-cn}\cdot \sum_{\alpha }{\hat{f}}^{2}\left(\alpha \right),$$

*where c is an absolute constant depending on ρ and μ.*


Let us remark that it seems helpful to have an explicit hypercontractive inequality (given by Theorem 1) from which both Corollaries 2 and 3 can be derived as special cases.

The following two results are simple consequences of Corollaries 2 and 3, respectively. Recall that for a subset $D\subseteq {\left\{0,1\right\}}^{n}$, λ(D) is the maximal eigenvalue of the adjacency matrix of the subgraph of the discrete cube induced by the vertices of *D*. Recall also that $R_{LP}\left(\delta \right)$ denotes the best possible upper bound on the asymptotic maximal rate R(δ) of a code with relative distance δ which is possible to obtain using the linear programming approach of [15].

**Corollary** **4.**

*Let D be a subset of ${\left\{0,1\right\}}^{n}$ of cardinality $\left|D\right|=2^{H(\rho )n}$, for some 0≤ρ≤1. Then*

$$\lambda \left(D\right)\;\leq \;2\sqrt{\rho (1-\rho )}\cdot n.$$


*This is almost tight if D is a Hamming ball of exponentially small cardinality.*

*For any $0\leq \delta \leq \frac{1}{2}$ holds*

$$R_{LP}\left(\delta \right)\;\geq \;\frac{1-H\left(\delta \right)+H\left(1/2-\sqrt{\delta (1-\delta )}\right)}{2}.$$




Some comments.

–As discussed above, the first of the these claims answers the question of [19] and shows that a certain approach to bound binary codes does not lead to an improvement of the first JPL bound. The second claim shows that the best possible bound obtainable via the linear programming approach of [15] is not better than the arithmetic average of the Gilbert-Varshamov bound and the first JPL bound. Observe that the first claim is a consequence of the logarithmic Sobolev inequality in Corollary 2, and hence of the behavior of the norm $\kappa_{2,2}$ in Theorem 1 as ϵ→0. The second claim is a consequence of the uncertainty principle in Corollary 3, and hence of the behavior of the norm $\kappa_{2,2}$ in Theorem 1 as ϵ∼0. We find these connections between notions to be rather intriguing.–As we have mentioned, the first of the claims recovers a result of [20], where it was also derived from the appropriate logarithmic Sobolev inequality. (Apart from this claim being a simple corollary of Theorem 1, an additional reason for stating it here is that it has only appeared in the unpublished arXiv preprint [20].) The second claim of recovers a result of [17].

Finally we present a result of a somewhat different nature. The question of the maximal possible ratio $\frac{{∥f∥}_{2}}{{∥f∥}_{1}}$ for a polynomial *f* of degree *s* on ${\left\{0,1\right\}}^{n}$ is considered in analysis [25,26] in connection with a conjecture of Pelczynski. The following claim is a simple consequence of Corollary 2.

**Corollary** **5.**
*Let $0\leq s\leq \frac{n}{2}$ and let f be a polynomial of degree s on ${\left\{0,1\right\}}^{n}$ (that is, f a restriction of a degree s polynomial on $R^{n}$ to ${\left\{0,1\right\}}^{n}$). Then, writing σ for $\frac{s}{n}$,*

$$\frac{1}{n}\log_{2}\left(\frac{{∥f∥}_{2}}{{∥f∥}_{1}}\right)\;\leq \;\frac{1-H\left(\frac{1}{2}-\sqrt{\sigma (1-\sigma )}\right)}{2}.$$



We remark that this improves the estimate of [25] for $0.3..\leq \frac{s}{n}<\frac{1}{2}$.

### 1.3. Related Work

In [10], it was shown that if $\frac{{∥f∥}_{p}}{{∥f∥}_{1}}\geq 2^{\rho n}$, for some p≥1 and ρ≥0, then ${∥f∥}_{p}\geq {∥f_{\epsilon }∥}_{1+\frac{p-1}{{\left(1-2\epsilon \right)}^{2}}+\Delta \left(p,\rho ,\epsilon \right)}$, where Δ(p,ρ,ϵ)>0 for all p>1, ϵ,ρ>0 (cf. with (Equation 1), which can be restated as ${∥f∥}_{p}\geq {∥f_{\epsilon }∥}_{1+\frac{p-1}{{\left(1-2\epsilon \right)}^{2}}}$, for $p=1+{\left(1-2\epsilon \right)}^{2}\left(q-1\right)$). The function Δ(p,ρ,ϵ) is “semi-explicit”, in the following sense: it is an explicit function of the (unique) solution of a certain explicit differential equation.

In [11], it was shown, using a different approach, that (restating the result in the notation of this paper) $∥f_{\epsilon }{∥}_{2}\leq {∥f∥}_{q}$, where *q* is determined by $F_{f,\epsilon }\left(q\right)=q$ (in the notation of the last comment above to Theorem 1). As we have observed, this is the best possible value for κ in Theorem 1, but it might not be easy to determine explicitly in practice (compare with Corollary 1).

In [27], Mrs. Gerber type inequalities for Rényi divergence and arbitrary distributions on Polish spaces were proved, using a different approach. The results in [27] apply in higher generality, but they seem to be somewhat less explicit than these in Proposition 1.

This paper is organized as follows. We prove Proposition 1 in Section 2 and Theorem 1 in Section 3. We prove the remaining claims, including some technical lemmas and claims made above in the comments to the main results, in Section 4.

## 2. Proof of Proposition 1

We first prove (Equation 4) and then show it to be tight. We prove (Equation 4) in two steps, using Theorem 2 to reduce it to a claim about properties of the function $\phi_{\epsilon }$, and then proving that claim.

We start with the first step. It follows closely the proof of Theorem 1.8 in [9], and hence will be presented rather briefly, and not in a self-contained manner. Let *f* be a function on ${\left\{0,1\right\}}^{n}$, for which we want to show (Equation 4). Recall that, by assumption, Ef=1. This means that ${∥f∥}_{\infty }\leq 2^{n}$, and that the points at which $f<2^{-n}$, say, contribute little to both sides ot (Equation 4), so we may ignore them for the sake of the discussion (that is, we may and will assume that *f* vanishes on these points). All the remaining points can be partitioned into O(n) level sets $A_{1},\ldots A_{r}$ such that *f* varies by a factor of 2 at most in each level set. Let $\alpha_{i}=\frac{1}{n}\log_{2}\left(|A_{i}|\right)$, and let $\nu_{i}=\frac{1}{n}\log_{2}\left(v_{i}\right)$, where $v_{i}$ is the minimal value of *f* on $A_{i}$. Then, as shown in the proof of Theorem 1.8 in [9], up to an additive error term of $O\left(\frac{\log(n)}{n}\right)$, we have,
$$\frac{Ent_{2}\left(f_{\epsilon }\right)}{n}\;=\;\frac{1}{n}\log_{2}{∥f_{\epsilon }∥}_{2}^{2}\;\leq \;2\cdot \max_{1\leq i\leq r}\left\{\phi_{\epsilon }\left(\alpha_{i}\right)+\nu_{i}\right\}.$$
The negligible error here contributes towards a negligible error in (Equation 4), which can then be removed by a tensorization argument, so we will ignore it from now on.

Let $N=\frac{1}{n}\log_{2}\left({∥f∥}_{q}\right)$. Note that $N=\frac{q-1}{q}\cdot \frac{Ent_{q}\left(f\right)}{n}$. Hence, in particular, $N\leq \frac{q-1}{q}$. Note also that for any 1≤i≤r holds $\alpha_{i}+\nu_{i}\leq 1$ (since Ef=1) and $\frac{\alpha_{i}-1}{q}+\nu_{i}\leq N$ (since $\frac{|A_{i}|}{2^{n}}2^{q\nu_{i}n}\leq \frac{1}{2^{n}}\sum_{x\in A_{i}}f^{q}\left(x\right)\leq {∥f∥}_{q}^{q}$). We also have $0\leq \alpha_{i}\leq 1$ and $-1\leq \nu_{i}\leq 1$. This discussion leads to the definition of the following two subsets of $R^{2}$, which will play an important role in the proof of Theorem 1 as well. (We remark that the relevance of the set Ω in the following definition is not immediately obvious. It will be made clear in the following arguments.)

**Definition** **2.**
*Let q>1 and $0<N\leq \frac{q-1}{q}$. Let $\Omega_{0}\subseteq R^{2}$ be defined by*

$$\Omega_{0}\;=\;\left\{\left(\alpha ,\nu \right):\;0\leq \alpha \leq 1,\;-1\leq \nu \leq 1,\;\alpha +\nu \leq 1,\;\frac{\alpha -1}{q}+\nu \leq N\right\}.$$

*Let $\Omega \subseteq \Omega_{0}$ be the set of all pairs $\left(\alpha ,\nu \right)\in \Omega_{0}$ with ν≥0.*


By the preceding discussion, (Equation 4) will follow from the following claim.

**Lemma** **2.**
*For all $0\leq \epsilon \leq \frac{1}{2}$ holds*

$$\max_{\left(\alpha ,\nu \right)\in \Omega_{0}}\left\{\phi_{\epsilon }\left(\alpha \right)+\nu \right\}\;=\;\frac{1}{2}\cdot \psi_{2,q}\left(\frac{qN}{q-1},\epsilon \right),$$

*where $\psi_{2,q}$ is defined in Definition 1.*


Before proving Lemma 2, we collect the relevant properties of the function $\phi_{\epsilon }$ in the following lemma.

**Lemma** **3.**
*Let $0<\epsilon <\frac{1}{2}$. Let $q_{0}=q_{0}\left(\epsilon \right)=1+{\left(1-2\epsilon \right)}^{2}$. The function $\phi_{\epsilon }$ has the following properties.*

*1.* 
*$\phi_{\epsilon }\left(\alpha \right)$ is strictly concave and increasing from $\phi_{\epsilon }\left(0\right)=-\frac{\log_{2}\left(\frac{4}{q_{0}}\right)}{2}$ to 0 on 0≤α≤1.*
*2.* 
*$\phi_{\epsilon }^{\prime }\left(0\right)=1$, $\phi_{\epsilon }^{\prime }\left(1\right)=\frac{1}{q_{0}}$.*
*3.* 
*$\frac{\alpha -1}{\phi_{\epsilon }\left(\alpha \right)}$ is strictly increasing in α, going up to $q_{0}$, as α→1.*
*4.* 
*The function $g\left(\alpha \right)=1-\alpha -\frac{\alpha }{1-\alpha }\cdot \phi_{\epsilon }\left(\alpha \right)$ is strictly decreasing on [0,1]. Moreover, g(0)=1 and $g\left(1\right)=\frac{1}{q_{0}}$.*



This lemma will be proved in Section 4. For now we assume its correctness, and proceed with the proof of Lemma 2.

**Proof.** Our first observation is that the maximum of $\phi_{\epsilon }\left(\alpha \right)+\nu$ on $\Omega_{0}$ is located in Ω, since for any point $\left(\alpha ,\nu \right)\in \Omega_{0}$ with ν<0, the point (α,0) is in Ω. So we may and will replace $\Omega_{0}$ with Ω in the following argument.Since $\phi_{\epsilon }$ is increasing, any local maximum of $\phi_{\epsilon }\left(\alpha \right)+\nu$ is located on the upper boundary of Ω, that is on the piecewise linear curve which starts as the straight line $\frac{\alpha }{q}+\nu =N+\frac{1}{q}$, for $0\leq \alpha \leq 1-\frac{qN}{q-1}$ and continues as the straight line α+ν=1 for $1-\frac{qN}{q-1}\leq \alpha \leq 1$.Note that, since $\phi_{\epsilon }^{\prime }<1$ for α>0, the function $\phi_{\epsilon }\left(\alpha \right)+\nu$ decreases (as a function of α) on the line α+ν=1 for $1-\frac{qN}{q-1}\leq \alpha \leq 1$. Next, let $h\left(\alpha \right)=\phi_{\epsilon }\left(\alpha \right)-\frac{\alpha }{q}+\left(N+\frac{1}{q}\right)$. The function *h* describes the restriction of $\phi_{\epsilon }\left(\alpha \right)+\nu$ to the line $\frac{\alpha }{q}+\nu =N+\frac{1}{q}$, and we are interested on the maximum of *h* on the interval $I=\left\{0\leq \alpha \leq 1-\frac{qN}{q-1}\right\}$. We have $h^{\prime }\left(\alpha \right)=\phi_{\epsilon }^{\prime }\left(\alpha \right)-\frac{1}{q}$. By Lemma 3, the function *h* is concave, and hence there are two possible cases:
$\phi_{\epsilon }^{\prime }\left(1-\frac{qN}{q-1}\right)\geq \frac{1}{q}$. In this case *h* is increasing on *I* and we get
$$\max_{(\alpha ,\nu )\in \Omega }\left\{\phi_{\epsilon }\left(\alpha \right)+\nu \right\}\;=\;\max_{\alpha \in I}\left\{h\left(\alpha \right)\right\}\;=\;h\left(1-\frac{qN}{q-1}\right)\;=\;$$
$$\phi_{\epsilon }\left(1-\frac{qN}{q-1}\right)+\frac{qN}{q-1}\;=\;\frac{1}{2}\cdot \psi_{2,q}\left(\frac{qN}{q-1},\epsilon \right).$$The last equality follows from the definition of $\psi_{2,q}$ in this case.$\phi_{\epsilon }^{\prime }\left(1-\frac{qN}{q-1}\right)<\frac{1}{q}$. Note that, by Lemma 3, $1=\phi_{\epsilon }^{\prime }\left(0\right)>\frac{1}{q}$. Hence, in this case the maximum of *h* on *I* is located at the unique zero of its derivative, that is at the point $\alpha_{0}$ such that $\phi_{\epsilon }^{\prime }\left(\alpha_{0}\right)=\frac{1}{q}$. Using the definition of $\psi_{2,q}$ in this case, we get
$$\max_{(\alpha ,\nu )\in \Omega }\left\{\phi_{\epsilon }\left(\alpha \right)+\nu \right\}\;=\;h\left(\alpha_{0}\right)\;=\;N+\left(\phi_{\epsilon }\left(\alpha_{0}\right)+\frac{1-\alpha_{0}}{q}\right)\;=\;\frac{1}{2}\cdot \psi_{2,q}\left(\frac{qN}{q-1},\epsilon \right).$$
□

This concludes the proof of (Equation 4). The fact that $\psi_{2,q}\left(x,\epsilon \right)$ is strictly increasing and concave in its first argument is an easy implication of Lemma 3.

We pass to showing the tightness of (Equation 4). Let $0<\epsilon <\frac{1}{2}$ and 0<x<1. Set $N=\frac{q-1}{q}\cdot x$. Let Ω be the domain defined in Definition 2, and let $\left(\alpha^{*},\nu^{*}\right)$ be the maximum point of $\phi_{\epsilon }\left(\alpha \right)+\nu$ on Ω (note that the discussion above determines this point uniquely). We proceed to define the function *f*. Let *n* be sufficiently large. For $y\in {\left\{0,1\right\}}^{n}$, let |y| denotes the Hamming weight of *y*, that is the number of 1-coordinates in *y*. Let $r=\lfloor H^{-1}\left(\alpha^{*}\right)\cdot n\rfloor$. Let $S=\{y\in {\left\{0,1\right\}}^{n},|y|=r\}$ be the Hamming sphere around zero of radius *r* in ${\left\{0,1\right\}}^{n}$. Now there are two cases to consider.

If $\phi_{\epsilon }^{\prime }\left(1-x\right)<\frac{1}{q}$, then by the discussion above, the point $\left(\alpha^{*},\nu^{*}\right)$ lies on the line $\frac{\alpha }{q}+\nu =N+\frac{1}{q}$, but not on the line α+ν=1. Observe that $2^{\alpha^{*}n-o\left(n\right)}\leq \left|S\right|\leq 2^{\alpha^{*}n}$ (the first estimate follows from the Stirling formula, for the second estimate see, e.g., Theorem 1.4.5. in [28]). As the first attempt, let $g=2^{\nu^{*}n}\cdot 1_{S}$. Then $N-o\left(n\right)\leq \frac{\alpha^{*}-1}{q}+\nu^{*}-o\left(n\right)\leq \frac{1}{n}\log_{2}{∥g∥}_{q}\leq \frac{\alpha^{*}-1}{q}+\nu^{*}=N$. That is, $x-o_{n}\left(1\right)\leq \frac{Ent_{q}\left(g\right)}{n}\leq x$. However, Eg is exponentially small. To correct that, we define *f* to be $v=2^{\left(\nu^{*}-\delta \right)\cdot n}$ on *S*, and $\frac{2^{n}-\left|S\right|v}{2^{n}-\left|S\right|}$ on the complement of *S*. Then Ef=1. We choose δ to be as small as possible, while ensuring that $\frac{Ent_{q}\left(f\right)}{n}\leq x$. Since the contribution of the constant-1 function to ${∥f∥}_{q}$ is exponentially small w.r.t. ${∥f∥}_{q}$, we can choose $\delta =o_{n}\left(1\right)$. We now have Ef=1, $\frac{Ent_{q}\left(f\right)}{n}\leq x$, and
$$\frac{Ent_{2}\left(f_{\epsilon }\right)}{n}\;=\;\frac{1}{n}\log_{2}{∥f_{\epsilon }∥}_{2}^{2}\;=\;\frac{1}{n}\log_{2}\langle f_{2\epsilon (1-\epsilon )},f\rangle \;\geq \;$$
$$2\cdot \left(\phi_{\epsilon }\left(\alpha^{*}\right)+\nu^{*}\right)-o_{n}\left(1\right)\;\geq \;\psi_{2,q}\left(x,\epsilon \right)-o_{n}\left(1\right).$$Here the second equality follows from the semigroup property of the noise operator: $T_{\epsilon }\circ T_{\epsilon }=T_{2\epsilon (1-\epsilon )}$. The first inequality follows from the tightness part of Theorem 2 and the definition of $\phi_{\epsilon }$. The second inequality follows from Lemma 2.The tightness of (Equation 4) in this case now follows, taking into account the fact that $\psi_{2,q}$ is strictly increasing.If $\phi_{\epsilon }^{\prime }\left(1-x\right)\geq \frac{1}{q}$, the point $\left(\alpha^{*},\nu^{*}\right)$ lies on the intersection of the lines $\frac{\alpha }{q}+\nu =N+\frac{1}{q}$, and α+ν=1. Hence the function $g=2^{\nu^{*}n}\cdot 1_{S}$ has both $x-o_{n}\left(1\right)\leq \frac{Ent_{q}\left(g\right)}{n}\leq x$, and $1-o_{n}\left(1\right)\leq Eg\leq 1$. It is easy to see that *g* can be corrected as in the preceding case, by decreasing it slightly on *S* and adding a constant component, to obtain a function *f* with expectation 1 and $Ent_{q}\left(f\right)\leq x$, and with $\frac{Ent_{2}\left(f_{\epsilon }\right)}{n}\geq \psi_{2,q}\left(x,\epsilon \right)-o_{n}\left(1\right)$, proving the tightness of (Equation 4) in this case as well. We omit the details.

This completes the proof of Proposition 1. □

## 3. Proof of Theorem 1

The high-level outline of the argument in this proof is similar to that of Proposition 1. We start with proving (Equation 5), doing this in two steps. In the first step Theorem 2 is used to reduce (Equation 5) to a claim about properties of the function $\phi_{\epsilon }$. That claim is proved in the second step.

We will give only a brief description of the first step since, similarly to the first step in the proof of Proposition 1, it follows closely the proof of Theorem 1.8 in [9]. Let *f* be a function on ${\left\{0,1\right\}}^{n}$, for which we may and will assume that $f\geq 2^{-n}$ and that $Ef={∥f∥}_{1}=1$. There are O(n) real numbers $0\leq \alpha_{1},...,\alpha_{r}\leq 1$ and $-1\leq \nu_{1},\ldots ,\nu_{r}\leq 1$, such that, up to a negligible error, which may be removed by tensorization, we have
$$\frac{1}{n}\log_{2}∥f_{\epsilon }{∥}_{2}\;\leq \;\max_{1\leq i\leq r}\left\{\phi_{\epsilon }\left(\alpha_{i}\right)+\nu_{i}\right\}\;\mathrm{and}\;\frac{1}{n}\log_{2}{∥f∥}_{q}\;=\;\max_{1\leq i\leq r}\left\{\frac{\alpha_{i}-1}{q}+\nu_{i}\right\}.$$

Hence (Equation 5) reduces to claim (Equation 6) in the following proposition.

**Proposition** **2.**
*Let q>1 and $0\leq \alpha_{1},\ldots ,\alpha_{r}\leq 1$, $-1\leq \nu_{1},\ldots ,\nu_{r}\leq 1$ with $\max_{1\leq i\leq r}\left\{\left(\alpha_{i}-1\right)+\nu_{i}\right\}=0$. Let $N=\max_{1\leq i\leq r}\left\{\frac{\alpha_{i}-1}{q}+\nu_{i}\right\}$. Then for any $0\leq \epsilon \leq \frac{1}{2}$ holds*

(6)
$$\max_{1\leq i\leq r}\left\{\phi_{\epsilon }\left(\alpha_{i}\right)+\nu_{i}\right\}\;\leq \;\max_{1\leq i\leq r}\left\{\frac{\alpha_{i}-1}{\kappa }+\nu_{i}\right\},$$

*where $\kappa =\kappa_{2,q}\left(\frac{qN}{q-1},\epsilon \right)$ is defined in Definition 1 (it is easy to see that $0\leq N\leq \frac{q-1}{q}$, and hence κ is well defined).*

*Moreover, this is tight, in the following sense. For any $0<N<\frac{q-1}{q}$ and $0<\epsilon <\frac{1}{2}$, and for any $\tilde{\kappa }<\kappa_{2,q}\left(x,\epsilon \right)$, there exist $0\leq \alpha_{1},\alpha_{2}\leq 1$ and $-1\leq \nu_{1},\nu_{2}\leq 1$ such that $\max_{1\leq i\leq 2}\left\{\left(\alpha_{i}-1\right)+\nu_{i}\right\}=0$, $\max_{1\leq i\leq 2}\left\{\frac{\alpha_{i}-1}{q}+\nu_{i}\right\}=N$, and $\max_{1\leq i\leq 2}\left\{\phi_{\epsilon }\left(\alpha_{i}\right)+\nu_{i}\right\}>\max_{1\leq i\leq r}\left\{\frac{\alpha_{i}-1}{\tilde{\kappa }}+\nu_{i}\right\}$.*


**Proof** **of** **Proposition** **2.**We start with verifying simple boundary cases. First, we observe that $\phi_{0}\left(x\right)=\frac{x-1}{2}$ (Lemma 9) and that $\phi_{\frac{1}{2}}\left(x\right)=x-1$ (see the relevant discussion in the proof of Corollary 1). In addition, it is easy to see that $\kappa_{2,q}\left(x,\frac{1}{2}\right)=1$ for all q≥1 and 0≤x≤1; and (bearing in mind that $\phi_{0}\left(x\right)=\frac{x-1}{2}$) that $\kappa_{2,q}\left(x,0\right)=2$ for all q≥1 and 0≤x≤1. Therefore (Equation 6) is an identity for ϵ=0 and $\epsilon =\frac{1}{2}$. Hence we may and will assume from now on that $0<\epsilon <\frac{1}{2}$.Let $q_{0}=1+{\left(1-2\epsilon \right)}^{2}$. We proceed to consider the (simple) cases N=0 or $N+\frac{1}{q}\leq \frac{1}{q_{0}}$. Note that in these cases we have $\kappa =\kappa_{2,q}\left(\frac{qN}{q-1},\epsilon \right)=q_{0}$. Next, observe that, by the first and the second claims of Lemma 3, for any 0≤α≤1 holds $\phi_{\epsilon }\left(\alpha \right)\leq \frac{\alpha -1}{q_{0}}=\frac{\alpha -1}{\kappa }$ and hence (Equation 6) holds trivially in these cases.We continue to prove (Equation 6), assuming from now on that N>0 and that $N+\frac{1}{q}>\frac{1}{q_{0}}$. Let $\Omega \subseteq R^{2}$ be the set defined in Definition 2. We now define a family of continuous functions on Ω, which will play an important role in the following argument. Let $\left(\alpha_{1},\nu_{1}\right)$ be a point in Ω with $\frac{\alpha_{1}-1}{q}+\nu_{1}=N$. Define a function $f=f_{\alpha_{1},\nu_{1}}$ on Ω as follows. For (α,ν)∈Ω with α<1 let f(α,ν) be the value of κ for which $\phi_{\epsilon }\left(\alpha \right)+\nu =\max\left\{\frac{\alpha_{1}-1}{\kappa }+\nu_{1},\frac{\alpha -1}{\kappa }+\nu \right\}$. In addition, let $f\left(1,0\right)=\frac{1-\alpha_{1}}{\nu_{1}}$.**Lemma** **4.**
*For any choice of $\left(\alpha_{1},\nu_{1}\right)$ as above the function $f_{\alpha_{1},\nu_{1}}$ is well-defined and continuous on *Ω*.*
Let $M\left(\alpha_{1},\nu_{1}\right)=\max_{\Omega }f_{\alpha_{1},\nu_{1}}$. The inequality (Equation 6) will follow from the next main technical claim, describing the behavior of $M\left(\alpha_{1},\nu_{1}\right)$, as a function of $\alpha_{1}$ and $\nu_{1}$. Before stating this claim, let us make some preliminary comments. Note that the points $\left(1-\frac{qN}{q-1},\frac{qN}{q-1}\right)$ and $\left(0,N+\frac{1}{q}\right)$ are possible choices for $\left(\alpha_{1},\nu_{1}\right)$. Note also that $\alpha_{0}$ in the third part of the claim is well-defined, by the fourth claim of Lemma 3.**Proposition** **3.**
*1.* 

$$M\left(1-\frac{qN}{q-1},\frac{qN}{q-1}\right)\;=\;\frac{-\frac{qN}{q-1}}{\phi_{\epsilon }\left(1-\frac{qN}{q-1}\right)}.$$

*2.* 
*If $\frac{-\frac{qN}{q-1}}{\phi_{\epsilon }\left(1-\frac{qN}{q-1}\right)}\geq q$, then for any choice of $\left(\alpha_{1},\nu_{1}\right)$ holds*

$$M\left(\alpha_{1},\nu_{1}\right)\;\leq \;M\left(1-\frac{qN}{q-1},\frac{qN}{q-1}\right).$$

*3.* 
*If $\frac{-\frac{qN}{q-1}}{\phi_{\epsilon }\left(1-\frac{qN}{q-1}\right)}\leq q$, then for any choice of $\left(\alpha_{1},\nu_{1}\right)$ holds*

$$M\left(1-\frac{qN}{q-1},\frac{qN}{q-1}\right)\;\leq \;M\left(\alpha_{1},\nu_{1}\right)\;\leq \;M\left(0,N+\frac{1}{q}\right)\;=\;\frac{\alpha_{0}-1}{\phi_{\epsilon }\left(\alpha_{0}\right)},$$


$$\mathrm{where}\;\;\alpha_{0}\;\;\mathrm{is}\;\mathrm{determined}\;\mathrm{by}\;\;1-\alpha_{0}-\frac{\alpha_{0}\phi_{\epsilon }\left(\alpha_{0}\right)}{1-\alpha_{0}}\;=\;N+\frac{1}{q}.$$


We will prove Lemma 4 and Proposition 3 in Section 3.1 and Section 3.2. For now we assume their validity and complete the proof of Proposition 2.We first prove (Equation 6). Note that if $x=\frac{qN}{q-1}$ then in the definition of $\kappa_{2,q}\left(x,\epsilon \right)$ we have $y=\frac{q-1}{q}\cdot x+\frac{1}{q}=N+\frac{1}{q}$. Recall also that we may assume that N>0 and that $y=N+\frac{1}{q}>\frac{1}{q_{0}}$.By assumption $\alpha_{i}+\nu_{i}\leq 1$, and $\frac{\alpha_{i}-1}{q}+\nu_{i}\leq N$ for all 1≤i≤r. Moreover there is an index 1≤i≤r for which $\frac{\alpha_{i}-1}{q}+\nu_{i}=N$. Assume, w.l.o.g., that i=1. We apply Proposition 3 to the function $f_{\alpha_{1},\nu_{1}}$. Observe that the claim of the proposition together with the definition of κ imply $M\left(\alpha_{1},\nu_{1}\right)\leq \kappa$. By the definition of $f_{\alpha_{1},\nu_{1}}$, this means that for any point $\left(\alpha ,\nu \right)\in \Omega$ holds $\phi_{\epsilon }\left(\alpha \right)+\nu \leq \max\left\{\frac{\alpha_{1}-1}{\kappa }+\nu_{1},\frac{\alpha -1}{\kappa }+\nu \right\}$. We now claim that this inequality holds for all the points $\left(\alpha_{i},\nu_{i}\right)$, 1≤i≤r, which will immediately imply (Equation 6). In fact, points $\left(\alpha_{i},\nu_{i}\right)$ with $0\leq \nu_{i}\leq 1$ lie in Ω and hence the inequality holds for these points. Furthermore, if $\nu_{i}<0$ for some 1≤i≤r, then the point $\left(\alpha_{i},0\right)$ lies in Ω, and hence $\phi_{\epsilon }\left(\alpha_{i}\right)\leq \max\left\{\frac{\alpha_{1}-1}{q}+\nu_{1},\frac{\alpha_{i}-1}{q}\right\}$. However, then $\phi_{\epsilon }\left(\alpha_{i}\right)+\nu_{i}\leq \max\left\{\frac{\alpha_{1}-1}{q}+\nu_{1},\frac{\alpha_{i}-1}{q}+\nu_{i}\right\}$, proving the inequality in this case as well.We pass to proving the tightness of (Equation 6), starting with the case $N+\frac{1}{q}\leq \frac{1}{q_{0}}$. In this case, by definition, $\kappa =q_{0}$. Let $\tilde{\kappa }<\kappa$ be given. Observe that since, by assumption, N>0, we have $q>q_{0}$. Set $\alpha_{1}=\frac{\frac{1}{q_{0}}-\frac{1}{q}-N}{\frac{1}{q_{0}}-\frac{1}{q}}$. Set $\nu_{1}=\frac{1-\alpha_{1}}{q_{0}}$. Let δ>0 be sufficiently small (depending on *N* and $\tilde{\kappa }$). Set $\alpha_{2}=1-\delta$ and $\nu_{2}=\delta$. It is easy to see that $\alpha_{1},\alpha_{2}$ and $\nu_{1}$, $\nu_{2}$ satisfy the required constraints. We claim that $\phi_{\epsilon }\left(\alpha_{2}\right)+\nu_{2}>\max_{1\leq i\leq 2}\left\{\frac{\alpha_{i}-1}{\tilde{\kappa }}+\nu_{i}\right\}$. In fact, for a sufficiently small δ we have, using the second claim of Lemma 3 (and observing that $\phi_{\epsilon }^{\prime }$ is continuous), that
$$\phi_{\epsilon }\left(\alpha_{2}\right)+\nu_{2}\;=\;\phi_{\epsilon }\left(1-\delta \right)+\delta \;\approx \;-\frac{\delta }{q_{0}}+\delta \;>\;-\frac{\delta }{\tilde{\kappa }}+\delta \;=\;\frac{\alpha_{2}-1}{\tilde{\kappa }}+\nu_{2},$$
and
$$\phi_{\epsilon }\left(\alpha_{2}\right)+\nu_{2}\;\approx \;-\frac{\delta }{q_{0}}+\delta \;>\;0\;\geq \;\frac{\alpha_{1}-1}{\tilde{\kappa }}+\frac{1-\alpha_{1}}{q_{0}}\;=\;\frac{\alpha_{1}-1}{\tilde{\kappa }}+\nu_{1}.$$We pass to the case $N+\frac{1}{q}>\frac{1}{q_{0}}$ and $\frac{-\frac{qN}{q-1}}{\phi_{\epsilon }\left(1-\frac{qN}{q-1}\right)}\geq q$. In this case $\kappa =\frac{-\frac{qN}{q-1}}{\phi_{\epsilon }\left(1-\frac{qN}{q-1}\right)}$. Set $\alpha_{1}=\alpha_{2}=1-\frac{qN}{q-1}$ and $\nu_{1}=\nu_{2}=\frac{qN}{q-1}$. It is easy to see that $\alpha_{1},\alpha_{2}$ and $\nu_{1}$, $\nu_{2}$ satisfy the required constraints. It is also easy to see that for any $\tilde{\kappa }<\kappa$ holds
$$\phi_{\epsilon }\left(\alpha_{1}\right)+\nu_{1}\;=\;\frac{\alpha_{1}-1}{\kappa }+\nu_{1}\;>\;\frac{\alpha_{1}-1}{\tilde{\kappa }}+\nu_{1}.$$It remains to deal with the case $N+\frac{1}{q}>\frac{1}{q_{0}}$ and $\frac{-\frac{qN}{q-1}}{\phi_{\epsilon }\left(1-\frac{qN}{q-1}\right)}<q$. Let $\alpha_{0}$ be determined by $1-\alpha_{0}-\frac{\alpha_{0}\phi_{\epsilon }\left(\alpha_{0}\right)}{1-\alpha_{0}}=N+\frac{1}{q}$. Then $\kappa =\frac{\alpha_{0}-1}{\phi_{\epsilon }\left(\alpha_{0}\right)}$. Set $\alpha_{1}=0$ and $\nu_{1}=N+\frac{1}{q}$. Set $\alpha_{2}=\alpha_{0}$ and $\nu_{2}=1-\alpha_{0}$. It is easy to see that in this case the function $1-\alpha -\frac{\alpha \phi_{\epsilon }\left(\alpha \right)}{1-\alpha }$ is larger than $N+\frac{1}{q}$ at $\alpha =1-\frac{qN}{q-1}$, and hence the fourth claim of Lemma 3 implies that $\alpha_{2}=\alpha_{0}>1-\frac{qN}{q-1}$. Using this, it is easy to see that $\alpha_{1},\alpha_{2}$ and $\nu_{1}$, $\nu_{2}$ satisfy the required constraints. Furthermore, note that $\alpha_{2}<1$ (again, using the fourth claim of Lemma 3). It is also easy to verify, using the definition of $\alpha_{0}$, that
$$\phi_{\epsilon }\left(\alpha_{2}\right)+\nu_{2}\;=\;\frac{\alpha_{1}-1}{\kappa }+\nu_{1}\;=\;\frac{\alpha_{2}-1}{\kappa }+\nu_{2},$$
which implies that for any $\tilde{\kappa }<\kappa$ holds $\phi_{\epsilon }\left(\alpha_{2}\right)+\nu_{2}>\max_{1\leq i\leq 2}\left\{\frac{\alpha_{i}-1}{\tilde{\kappa }}+\nu_{i}\right\}$. This completes the proof of Proposition 2. □

We now prove Lemma 4 and Proposition 3. Recall that we may assume N>0 and $N+\frac{1}{q}>\frac{1}{q_{0}}$.

### 3.1. Proof of Lemma 4

Let $\left(\alpha_{1},\nu_{1}\right)$ be a point in Ω with $\frac{\alpha_{1}-1}{q}+\nu_{1}=N$. We start with some simple but useful observations about $\alpha_{1}$ and $\nu_{1}$.

**Lemma** **5.**
*1.* 
*$\alpha_{1}\leq 1-\frac{qN}{q-1}$ and $\nu_{1}\geq \frac{qN}{q-1}$.*
*2.* 
*$\frac{1-\alpha_{1}}{\nu_{1}}<q_{0}$.*



**Proof.** The first claim of the lemma is an easy consequence of the inequalities $\frac{\alpha_{1}-1}{q}+\nu_{1}=N$ and $\alpha_{1}+\nu_{1}\leq 1$. We omit the details.We pass to the second claim of the lemma, distinguishing two cases, $q\leq q_{0}$ and $q>q_{0}$. If $q\leq q_{0}$, then $\nu_{1}=N+\frac{1-\alpha_{1}}{q}>\frac{1-\alpha_{1}}{q}\geq \frac{1-\alpha_{1}}{q_{0}}$. If $q>q_{0}$, we use the fact that $N+\frac{1}{q}>\frac{1}{q_{0}}$ to obtain $\frac{1-\alpha_{1}}{q_{0}}<\left(1-\alpha_{1}\right)\left(N+\frac{1}{q}\right)=\left(1-\alpha_{1}\right)\left(\frac{\alpha_{1}}{q}+\nu_{1}\right)$. Viewing the last expression as a function of $\alpha_{1}$, it is easy to see that it equals $\nu_{1}$ at $\alpha_{1}=0$ and that it decreases in $\alpha_{1}$. Hence $\nu_{1}\geq \left(1-\alpha_{1}\right)\left(\frac{\alpha_{1}}{q}+\nu_{1}\right)>\frac{1-\alpha_{1}}{q_{0}}$, completing the argument in this case as well. □

We now show that the function $f=f_{\alpha_{1},\nu_{1}}$ is well-defined and that its values lie in the interval $\left(0,q_{0}\right)$. By Lemma 5, $\alpha_{1}<1$ and $0<f\left(1,0\right)=\frac{1-\alpha_{1}}{\nu_{1}}<q_{0}$. Let now α<1. In this case the function $g\left(\kappa \right)=\max\left\{\frac{\alpha_{1}-1}{\kappa }+\nu_{1},\frac{\alpha -1}{\kappa }+\nu \right\}$ is a strictly increasing continuous function of κ, which is −∞ at κ=0. Furthermore, by Lemma 3, $\phi_{\epsilon }\left(\alpha \right)<\frac{\alpha -1}{q_{0}}$, implying that $g\left(q_{0}\right)>\phi_{\epsilon }\left(\alpha \right)+\nu$. Hence, by the intermediate value theorem, there exists a unique $0<\kappa <q_{0}$ for which $\phi_{\epsilon }\left(\alpha \right)+\nu =\max\left\{\frac{\alpha_{1}-1}{\kappa }+\nu_{1},\frac{\alpha -1}{\kappa }+\nu \right\}$.

Next, we argue that *f* is continuous on Ω. Let $\left(\alpha ,\nu \right)\in \Omega$. If α<1, then there exists a compact neighborhood of $\left(\alpha ,\nu \right)$ in which both one-sided derivatives of g(κ) are positive and bounded. This, together with the fact that $\phi_{\epsilon }\left(\alpha \right)+\nu$ is continuous, implies that *f* is continuous at $\left(\alpha ,\nu \right)$.

It remains to argue that *f* is continuous at (1,0). Let *O* be a sufficiently small neighbourhood of (1,0) in Ω. Let $\left(\alpha ,\nu \right)\in O$, with α<1. Then $\phi_{\epsilon }\left(\alpha \right)+\nu$ is close to $\phi_{\epsilon }\left(1\right)+0=0$. We would like to claim that f(α,ν) is close to $f\left(1,0\right)=\frac{1-\alpha_{1}}{\nu_{1}}$. In fact, assume towards contradiction that f(α,ν) is significantly larger than $\frac{1-\alpha_{1}}{\nu_{1}}$. In this case $\phi_{\epsilon }\left(\alpha \right)+\nu =\max\left\{\frac{\alpha_{1}-1}{f(\alpha ,\nu )}+\nu_{1},\frac{\alpha -1}{f(\alpha ,\nu )}+\nu \right\}\geq \frac{\alpha_{1}-1}{f(\alpha ,\nu )}+\nu_{1}$ is significantly larger than 0 (taking into account that $\alpha_{1}<1$), reaching contradiction. On the other hand, assume that f(α,ν) is significantly smaller than $\frac{1-\alpha_{1}}{\nu_{1}}$, and hence significantly smaller than $q_{0}$ (by the second claim of Lemma 5). Recall that $\phi_{\epsilon }\left(1\right)=0$ and that $\phi_{\epsilon }^{\prime }\left(1\right)=\frac{1}{q_{0}}$. Hence $\phi_{\epsilon }\left(\alpha \right)=\frac{\alpha -1}{q_{0}}+O\left({\left(1-\alpha \right)}^{2}\right)>\frac{\alpha -1}{f(\alpha ,\nu )}$. This means that $\phi_{\epsilon }\left(\alpha \right)+\nu =\frac{\alpha_{1}-1}{f(\alpha ,\nu )}+\nu_{1}$, which is significantly smaller than 0, again reaching contradiction. This completes the proof of Lemma 4.

We collect some useful properties of $f=f_{\alpha_{1},\nu_{1}}$ in the following claim.

**Corollary** **6.**
*1.* 
*For any (α,ν)∈Ω holds $\phi_{\epsilon }\left(\alpha \right)+\nu =\max\left\{\frac{\alpha_{1}-1}{f(\alpha ,\nu )}+\nu_{1},\frac{\alpha -1}{f(\alpha ,\nu )}+\nu \right\}$.*
*2.* 
*$0<f\leq M\left(\alpha_{1},\nu_{1}\right)<q_{0}$ on *Ω*.*
*3.* 
*For any (α,ν)∈Ω holds $f\left(\alpha ,\nu \right)\leq \frac{\alpha -1}{\phi_{\epsilon }\left(\alpha \right)}$. (If α=1 we replace the RHS of this inequality with $q_{0}$.)*



**Proof.** The first two claims follow immediately from the preceding discussion and from the continuity of *f*. For the third claim, recall that
$$\phi_{\epsilon }\left(\alpha \right)+\nu =\max\left\{\frac{\alpha_{1}-1}{f(\alpha ,\nu )}+\nu_{1},\frac{\alpha -1}{f(\alpha ,\nu )}+\nu \right\}\geq \frac{\alpha -1}{f(\alpha ,\nu )}+\nu$$□

### 3.2. Proof of Proposition 3

Let $\left(\alpha_{1},\nu_{1}\right)$ be given, let $f=f_{\alpha_{1},\nu_{1}}$, and let $M=M\left(\alpha_{1},\nu_{1}\right)=\max_{\Omega }f$. Let $\left(\alpha^{*},\nu^{*}\right)$ be a maximum point of *f*. Then $f\left(\alpha^{*},\nu^{*}\right)=M$ and hence $\phi_{\epsilon }\left(\alpha^{*}\right)+\nu^{*}=\max\left\{\frac{\alpha_{1}-1}{M}+\nu_{1},\frac{\alpha^{*}-1}{M}+\nu \right\}$. Clearly either $\frac{\alpha_{1}-1}{M}+\nu_{1}\neq \frac{\alpha^{*}-1}{M}+\nu^{*}$ or $\frac{\alpha_{1}-1}{M}+\nu_{1}=\frac{\alpha^{*}-1}{M}+\nu^{*}$. In the first case we say that $\left(\alpha^{*},\nu^{*}\right)$ is a maximum point of the *first type*, and otherwise it is a maximum point of the *second type*.

The following two claims constitute the main steps of the proof of Proposition 3. They describe the respective behavior of maxima points of the first and the second type.

**Lemma** **6.**
*Let $\;\left(\alpha^{*},\nu^{*}\right)$ be a maximum point of f of the first type. Then the following two claims hold.*


*$\frac{\alpha_{1}-1}{f\left(\alpha^{*},\nu^{*}\right)}+\nu_{1}>\frac{\alpha^{*}-1}{f\left(\alpha^{*},\nu^{*}\right)}+\nu^{*}$.*

*$\alpha^{*}\leq 1-\frac{qN}{q-1}$.*



**Lemma** **7.**
*If $\left(\alpha_{1},\nu_{1}\right)=\left(1-\frac{qN}{q-1},\frac{qN}{q-1}\right)$, then $\left(1-\frac{qN}{q-1},\frac{qN}{q-1}\right)$ is the unique maximum point of f. This is a maximum point of the second type.*

*If $\left(\alpha_{1},\nu_{1}\right)\neq \left(1-\frac{qN}{q-1},\frac{qN}{q-1}\right)$, then there are two possible cases.*


*$\frac{-\frac{qN}{q-1}}{\phi_{\epsilon }\left(1-\frac{qN}{q-1}\right)}\geq q$. Let $\;\left(\alpha^{*},\nu^{*}\right)$ be a maximum point of f of the second type in this case. Then $\alpha^{*}\leq 1-\frac{qN}{q-1}$.*

*$\frac{-\frac{qN}{q-1}}{\phi_{\epsilon }\left(1-\frac{qN}{q-1}\right)}<q$. In this case f has a unique maximum point $\;\left(\alpha^{*},\nu^{*}\right)$. This point is of the second type. Furthermore, $\alpha^{*}>1-\frac{qN}{q-1}$, and it is uniquely determined by the following identity:*

$$\frac{\alpha^{*}-1}{\phi_{\epsilon }\left(\alpha^{*}\right)}\;=\;\frac{\alpha^{*}-\alpha_{1}}{\alpha^{*}-\left(1-\nu_{1}\right)}.$$




Lemmas 6 and 7 will be proved in Section 3.3. At this point we prove Proposition 3 assuming these lemmas hold.

We start with the first claim of Proposition 3. Let $\alpha_{1}=1-\frac{qN}{q-1}$ and $\nu_{1}=\frac{qN}{q-1}$. Let $f=f_{\alpha_{1},\nu_{1}}$. By the first claim of Lemma 7, we have
$$M\left(\alpha_{1},\nu_{1}\right)\;=\;f\left(\alpha_{1},\nu_{1}\right)\;=\;\frac{\alpha_{1}-1}{\phi_{\epsilon }\left(\alpha_{1},\nu_{1}\right)}\;=\;\frac{-\frac{qN}{q-1}}{\phi_{\epsilon }\left(1-\frac{qN}{q-1}\right)}.$$

We pass to the second claim of the proposition. Assume that $\frac{-\frac{qN}{q-1}}{\phi_{\epsilon }\left(1-\frac{qN}{q-1}\right)}\geq q$. Let $f=f_{\alpha_{1},\nu_{1}}$, for some $\alpha_{1}$ and $\nu_{1}$. Let $\left(\alpha^{*},\nu^{*}\right)$ be a maximum point of *f*. Then Lemmas 6 and 7 imply that $\alpha^{*}\leq 1-\frac{qN}{q-1}$. Hence
$$M\left(\alpha_{1},\nu_{1}\right)\;=\;f\left(\alpha^{*},\nu^{*}\right)\;\leq \;\frac{\alpha^{*}-1}{\phi_{\epsilon }\left(\alpha^{*}\right)}\;\leq \;\frac{-\frac{qN}{q-1}}{\phi_{\epsilon }\left(1-\frac{qN}{q-1}\right)}\;=\;M\left(1-\frac{qN}{q-1},\frac{qN}{q-1}\right).$$
Here in the second step we have used the third claim of Corollary 6, in the third step the third claim of Lemma 3 and in the fourth step the first claim of the proposition.

We pass to the third claim of the proposition. Assume that $\frac{-\frac{qN}{q-1}}{\phi_{\epsilon }\left(1-\frac{qN}{q-1}\right)}<q$. Let $f=f_{\alpha_{1},\nu_{1}}$, for some $\alpha_{1}$ and $\nu_{1}$. Then, by Lemma 7, *f* has a unique maximum point $\left(\alpha^{*},\nu^{*}\right)$. This means that $\alpha^{*}$ is determined by $\alpha_{1}$ and $\nu_{1}$, and furthermore, since $\nu_{1}=N+\frac{1-\alpha_{1}}{q}$, $\alpha^{*}$ is a function of $\alpha_{1}$. We will show the following claim below.

**Lemma** **8.**
*If $\left(\alpha_{1},\nu_{1}\right)\neq \left(1-\frac{qN}{q-1},\frac{qN}{q-1}\right)$ and $\frac{-\frac{qN}{q-1}}{\phi_{\epsilon }\left(1-\frac{qN}{q-1}\right)}<q$, then $\alpha^{*}$ is a decreasing function of $\alpha_{1}$.*


Assume Lemma 8 to hold. We have
$$M\left(\alpha_{1},\nu_{1}\right)\;=\;f\left(\alpha^{*},\nu^{*}\right)\;=\;\frac{\alpha^{*}\left(\alpha_{1}\right)-1}{\phi_{\epsilon }\left(\alpha^{*}\left(\alpha_{1}\right)\right)}\;\leq \;\frac{\alpha^{*}\left(0\right)-1}{\phi_{\epsilon }\left(\alpha^{*}\left(0\right)\right)}\;=\;M\left(0,N+\frac{1}{q}\right).$$
The second step uses the fact that $\left(\alpha^{*},\nu^{*}\right)$ is a maximum point of the second type, and hence $f\left(\alpha^{*},\nu^{*}\right)=\frac{\alpha^{*}-1}{\phi \left(\alpha^{*}\right)}$. The third step uses Lemma 8 and the third claim of Lemma 3, and the fourth step the fact that $\alpha_{1}=0$ implies $\nu_{1}=N+\frac{1}{q}$.

Next, by Lemma 7, $\alpha =\alpha^{*}\left(0\right)$ is determined by the identity $\frac{\alpha -1}{\phi_{\epsilon }\left(\alpha \right)}\;=\;\frac{\alpha }{\alpha -\left(\frac{q-1}{q}-N\right)}$ which, after rearranging, gives $1-\alpha -\frac{\alpha \phi_{\epsilon }\left(\alpha \right)}{1-\alpha }\;=\;N+\frac{1}{q}$. Hence, by the fourth claim of Lemma 3, $\alpha^{*}\left(0\right)=\alpha_{0}$ and $M\left(0,N+\frac{1}{q}\right)=\frac{\alpha_{0}-1}{\phi_{\epsilon }\left(\alpha_{0}\right)}$.

To conclude the proof of the third claim of the proposition, observe that since $\alpha^{*}>1-\frac{qN}{q-1}$, we have
$$M\left(\alpha_{1},\nu_{1}\right)\;=\;f\left(\alpha^{*},\nu^{*}\right)\;=\;\frac{\alpha^{*}-1}{\phi_{\epsilon }\left(\alpha^{*}\right)}\;>\;\frac{-\frac{qN}{q-1}}{\phi_{\epsilon }\left(1-\frac{qN}{q-1}\right)},$$
where the last inequality is by the third claim of Lemma 3. This completes the proof of Proposition 3.

It remains to prove Lemmas 6–8.

### 3.3. Proofs of the Remaining Lemmas

**Proof** **of** **Lemma** **6.**We start with the first claim of the lemma. Assume towards contradiction that $\frac{\alpha_{1}-1}{f\left(\alpha^{*},\nu^{*}\right)}+\nu_{1}<\frac{\alpha^{*}-1}{f\left(\alpha^{*},\nu^{*}\right)}+\nu^{*}$. Since *f* is a positive continuous function on Ω, there is a neighborhood *O* of $\left(\alpha^{*},\nu^{*}\right)$ in Ω on which $\frac{\alpha_{1}-1}{f\left(\alpha ,\nu \right)}+\nu_{1}<\frac{\alpha -1}{f\left(\alpha ,\nu \right)}+\nu$. This means that any point (α,ν)∈O satisfies $\phi_{\epsilon }\left(\alpha \right)+\nu =\frac{\alpha -1}{f\left(\alpha ,\nu \right)}+\nu$, and hence $f\left(\alpha ,\nu \right)=\frac{\alpha -1}{\phi_{\epsilon }\left(\alpha \right)}$. Since $f\left(\alpha^{*},\nu^{*}\right)\geq f\left(\alpha ,\nu \right)$, this implies that $\frac{\alpha^{*}-1}{\phi_{\epsilon }\left(\alpha^{*}\right)}\geq \frac{\alpha -1}{\phi_{\epsilon }\left(\alpha \right)}$, and hence, by the third claim of Lemma 3, that $\alpha^{*}\geq \alpha$. It follows that $\alpha^{*}$ has to be 1, and hence $\left(\alpha^{*},\nu^{*}\right)=\left(1,0\right)$. However, in this case $\frac{\alpha_{1}-1}{f\left(\alpha^{*},\nu^{*}\right)}+\nu_{1}=\frac{\alpha^{*}-1}{f\left(\alpha^{*},\nu^{*}\right)}+\nu^{*}=0$, reaching contradiction.We pass to the second claim of the lemma. By the first claim $\frac{\alpha_{1}-1}{f\left(\alpha^{*},\nu^{*}\right)}+\nu_{1}>\frac{\alpha^{*}-1}{f\left(\alpha^{*},\nu^{*}\right)}+\nu^{*}$. We claim that this implies that $\left(\alpha^{*},\nu^{*}\right)$ is a local maximum of $\phi_{\epsilon }\left(\alpha \right)+\nu$. In fact, arguing as above, there is a neighborhood *O* of $\left(\alpha^{*},\nu^{*}\right)$ on which $\frac{\alpha_{1}-1}{f\left(\alpha ,\nu \right)}+\nu_{1}>\frac{\alpha -1}{f\left(\alpha ,\nu \right)}+\nu$. This means that for any point (α,ν)∈O we have $\phi_{\epsilon }\left(\alpha \right)+\nu =\frac{\alpha_{1}-1}{f\left(\alpha ,\nu \right)}+\nu_{1}$. Since $f\left(\alpha^{*},\nu^{*}\right)\geq f\left(\alpha ,\nu \right)$, this implies that $\phi_{\epsilon }\left(\alpha \right)+\nu \leq \phi \left(\alpha^{*}\right)+\nu^{*}$. To complete the proof, recall that any local maximum (α,ν) of ϕ(α)+ν has $\alpha \leq 1-\frac{qN}{q-1}$ (as shown in the proof of Proposition 1). □

**Proof** **of** **Lemma** **7.**Let $\left(\alpha^{*},\nu^{*}\right)$ be a maximum point of *f* of the second type. The first observation is that $\left(\alpha^{*},\nu^{*}\right)$ has to lie on the upper boundary of Ω. In fact, assume not. Then for a sufficiently small τ>0 the point $\left(\alpha ,\nu \right)=\left(\alpha^{*},\nu^{*}+\tau \right)$ is in Ω. Since $f\left(\alpha ,\nu \right)\leq f\left(\alpha^{*},\nu^{*}\right)$, we have $\phi_{\epsilon }\left(\alpha \right)+\nu >\phi_{\epsilon }\left(\alpha^{*}\right)+\nu^{*}=\frac{\alpha_{1}-1}{f\left(\alpha^{*},\nu^{*}\right)}+\nu_{1}\geq \frac{\alpha_{1}-1}{f\left(\alpha ,\nu \right)}+\nu_{1}$. Hence $f\left(\alpha ,\nu \right)$ is determined by the equality $\phi_{\epsilon }\left(\alpha \right)+\nu =\frac{\alpha -1}{f\left(\alpha ,\nu \right)}+\nu$, which implies $f\left(\alpha ,\nu \right)=f\left(\alpha^{*},\nu^{*}\right)=\frac{\alpha^{*}-1}{\phi_{\epsilon }\left(\alpha^{*}\right)}$. Hence $\left(\alpha ,\nu \right)$ is a point of maximum of *f* of the first type with $\frac{\alpha_{1}-1}{f\left(\alpha ,\nu \right)}+\nu_{1}<\frac{\alpha -1}{f\left(\alpha ,\nu \right)}+\nu$. This, however, contradicts the first claim of Lemma 6.Recall that the upper boundary of Ω is a piecewise linear curve which starts as the straight line $\frac{\alpha }{q}+\nu =N+\frac{1}{q}$, for $0\leq \alpha \leq 1-\frac{qN}{q-1}$ and continues as the straight line α+ν=1 for $1-\frac{qN}{q-1}\leq \alpha \leq 1$. Hence there are two cases to consider: In the first case $\alpha^{*}\leq 1-\frac{qN}{q-1}$ and $\frac{\alpha^{*}}{q}+\nu^{*}=N+\frac{1}{q}$. In the second case $1-\frac{qN}{q-1}<\alpha^{*}\leq 1$ and $\alpha^{*}+\nu^{*}=1$.Assume that the second case holds. Then $\left(\alpha^{*},\nu^{*}\right)$ satisfies
$\frac{\alpha_{1}-1}{f\left(\alpha^{*},\nu^{*}\right)}+\nu_{1}=\frac{\alpha^{*}-1}{f\left(\alpha^{*},\nu^{*}\right)}+\nu^{*}=\phi_{\epsilon }\left(\alpha^{*}\right)+\nu^{*}$.$1-\frac{qN}{q-1}<\alpha^{*}\leq 1$ and $\alpha^{*}+\nu^{*}=1$.
In particular, $f\left(\alpha^{*},\nu^{*}\right)=\frac{\alpha^{*}-1}{\phi_{\epsilon }\left(\alpha^{*}\right)}=\frac{\alpha^{*}-\alpha_{1}}{\alpha^{*}-\left(1-\nu_{1}\right)}$. Consider the following two functions of α: $g_{1}\left(\alpha \right)=\frac{\alpha -1}{\phi_{\epsilon }\left(\alpha \right)}$ and $g_{2}\left(\alpha \right)=\frac{\alpha -\alpha_{1}}{\alpha -\left(1-\nu_{1}\right)}$, for $\alpha >1-\frac{qN}{q-1}$. Note that $g_{2}$ is well-defined since, by Lemma 5, $\nu_{1}\geq \frac{qN}{q-1}$. By the third claim of Lemma 3, $g_{1}$ is strictly increasing. On the other hand, $g_{2}\left(\alpha \right)=1+\frac{1-\alpha_{1}-\nu_{1}}{\alpha -\left(1-\nu_{1}\right)}$ is non-increasing. Note also that $g_{1}\left(1\right)=q_{0}$ (more precisely, $\lim_{\alpha \to 1}g_{1}\left(\alpha \right)=q_{0}$) and, by Lemma 5, $g_{2}\left(1\right)=\frac{1-\alpha_{1}}{\nu_{1}}<q_{0}$. This means that $g_{1}$ and $g_{2}$ coincide at a (unique) point $1-\frac{qN}{q-1}<\alpha <1$ iff $g_{1}\left(1-\frac{qN}{q-1}\right)<g_{2}\left(1-\frac{qN}{q-1}\right)$.Observe that if $\left(\alpha_{1},\nu_{1}\right)=\left(1-\frac{qN}{q-1},\frac{qN}{q-1}\right)$ then $g_{2}$ is the constant 1-function. Furthermore, by the first and the third claims of Lemma 3, $g_{1}\left(1-\frac{qN}{q-1}\right)\geq g_{1}\left(0\right)=\frac{2}{\log_{2}\left(4/q_{0}\right)}\geq 1$, and hence in this case $g_{1}$ and $g_{2}$ cannot coincide for $\alpha >1-\frac{qN}{q-1}$. If $\left(\alpha_{1},\nu_{1}\right)\neq \left(1-\frac{qN}{q-1},\frac{qN}{q-1}\right)$ then it is easy to see (recall that $\frac{\alpha_{1}}{q}+\nu_{1}=N+\frac{1}{q}$) that $g_{2}\left(1-\frac{qN}{q-1}\right)=q$, and hence the two functions have a unique intersection at some $\alpha >1-\frac{qN}{q-1}$ iff $g_{1}\left(1-\frac{qN}{q-1}\right)=\frac{-\frac{qN}{q-1}}{\phi_{\epsilon }\left(1-\frac{qN}{q-1}\right)}$ is smaller than *q*.To recap, the second case can hold only provided $\left(\alpha_{1},\nu_{1}\right)\neq \left(1-\frac{qN}{q-1},\frac{qN}{q-1}\right)$ and $\frac{-\frac{qN}{q-1}}{\phi_{\epsilon }\left(1-\frac{qN}{q-1}\right)}<q$. Furthermore, if it holds then $1-\frac{qN}{q-1}<\alpha^{*}<1$ is uniquely determined by the equality $g_{1}\left(\alpha^{*}\right)=g_{2}\left(\alpha^{*}\right)$.We can now complete the proof of the lemma. First, let $\left(\alpha_{1},\nu_{1}\right)=\left(1-\frac{qN}{q-1},\frac{qN}{q-1}\right)$. By the preceding discussion, in this case a maximum point $\left(\alpha^{*},\nu^{*}\right)$ of *f* of the second type has to have $\alpha^{*}\leq \alpha_{1}$. Moreover, taking into account Lemma 6, this is true for any maximum point of *f*. By the third claim of Corollary 6, this means that $M\left(\alpha_{1},\nu_{1}\right)\leq \frac{\alpha_{1}-1}{\phi_{\epsilon }\left(\alpha_{1}\right)}=f\left(\alpha_{1},\nu_{1}\right)$. Hence $\left(\alpha_{1},\nu_{1}\right)$ is a maximum point of *f*. It is trivially a maximum point of the second type. To see that it is a unique maximum point, note that for any point (α,ν) on the upper boundary of Ω, if $\alpha =\alpha_{1}$, then necessarily $\nu =\nu_{1}$. So, for any other putative maximum point (α,ν), we would have $\alpha <\alpha_{1}$ and hence, by the third claims of Lemma 3 and the third claim of Corollary 6, $f\left(\alpha ,\nu \right)\leq \frac{\alpha -1}{\phi_{\epsilon }\left(\alpha \right)}<\frac{\alpha_{1}-1}{\phi_{\epsilon }\left(\alpha_{1}\right)}=f\left(\alpha_{1},\nu_{1}\right)$. This proves the first claim of the lemma.Assume now that $\left(\alpha_{1},\nu_{1}\right)\neq \left(1-\frac{qN}{q-1},\frac{qN}{q-1}\right)$. Let $\left(\alpha^{*},\nu^{*}\right)$ be a maximum point of *f* of the second type. If $g_{1}\left(1-\frac{qN}{q-1}\right)=\frac{-\frac{qN}{q-1}}{\phi_{\epsilon }\left(1-\frac{qN}{q-1}\right)}\geq q$, then the preceding discussion implies that $\alpha^{*}\leq 1-\frac{qN}{q-1}$, proving the second claim of the lemma.If $\frac{-\frac{qN}{q-1}}{\phi_{\epsilon }\left(1-\frac{qN}{q-1}\right)}<q$, let α be the unique solution for $g_{1}\left(\alpha \right)=g_{2}\left(\alpha \right)$ on $1-\frac{qN}{q-1}<\alpha <1$. Set $\alpha^{*}=\alpha$ and $\nu^{*}=1-\alpha$. We claim that $\left(\alpha^{*},\nu^{*}\right)$ is the unique maximum point of *f* (note that by Lemma 6 it would necessarily be of the second type). In fact, let us first verify that $\frac{\alpha_{1}-1}{\kappa }+\nu_{1}=\frac{\alpha^{*}-1}{\kappa }+\nu^{*}=\phi_{\epsilon }\left(\alpha^{*}\right)+\nu^{*}$, for $\kappa =\frac{\alpha^{*}-1}{\phi_{\epsilon }\left(\alpha^{*}\right)}$. The second equality is immediate, by the definition of κ. The first equality is equivalent to $\kappa =\frac{\alpha^{*}-\alpha_{1}}{\alpha^{*}-\left(1-\nu_{1}\right)}$, which follows from the definitions of $\alpha^{*}$ and κ. Hence $f\left(\alpha^{*},\nu^{*}\right)=\kappa =\frac{\alpha^{*}-1}{\phi_{\epsilon }\left(\alpha^{*}\right)}$. For any other putative maximum point (α,ν), we would have, by the preceding discussion, that $\alpha \leq 1-\frac{qN}{q-1}<\alpha^{*}$ and hence, as above, $f\left(\alpha ,\nu \right)\leq \frac{\alpha -1}{\phi_{\epsilon }\left(\alpha \right)}<f\left(\alpha^{*},\nu^{*}\right)$. This proves the third claim of the lemma. □

**Proof** **of** **Lemma** **8.**In the assumptions of the lemma, $\alpha^{*}$ is the unique solution on $\left(1-\frac{qN}{q-1},1\right)$ of the identity
$$\frac{\alpha^{*}-1}{\phi_{\epsilon }\left(\alpha^{*}\right)}\;=\;\frac{\alpha^{*}-\alpha_{1}}{\alpha^{*}-\left(1-\nu_{1}\right)}.$$
Here the LHS is a strictly increasing and the RHS a strictly decreasing (since by assumption $\alpha_{1}\neq 1-\frac{qN}{q-1}$, and hence $\alpha_{1}+\nu_{1}<1$) functions of $\alpha^{*}$. It follows that to prove the claim of the lemma it suffices to show that for a fixed $\alpha^{*}>1-\frac{qN}{q-1}$ the RHS is a decreasing function of $\alpha_{1}$ (keeping in mind that $\nu_{1}=-\frac{\alpha_{1}}{q}+\left(N+\frac{1}{q}\right)$). However, this is easily verifiable by a direct differentiation of the RHS w.r.t. $\alpha_{1}$. □

This completes the proof of Proposition 2 and of (Equation 5). We proceed to complete the proof of Theorem 1. The tightness of (Equation 5) follows from the tightness of (Equation 6), similarly to the way the tightness of (Equation 4) was shown in the proof of Proposition 1. We omit the details.

It remains to consider the properties of the function $\kappa_{2,q}$. We first remark that it is easy to see, using the properties of the function $\phi_{\epsilon }$ given in Lemma 3, that $\kappa_{2,q}$ is a continuous function of its first variable (we omit the details). In particular, we can replace strict inequalities with non-strict ones in the definition of $\kappa_{2,q}$ in Definition 1. Now there are two cases to consider.

$q\geq q_{0}$. In this case, by the third claim of Lemma 3, $-\frac{x}{\phi_{\epsilon }\left(1-x\right)}$ is never larger than *q*, and hence
$$\kappa_{2,q}\left(x,\epsilon \right)\;=\;\left\{\begin{array}{ccc} q_{0} & if & y\leq \frac{1}{q_{0}}\\ \frac{\alpha_{0}-1}{\phi_{\epsilon }\left(\alpha_{0}\right)} & if & y\geq \frac{1}{q_{0}} \end{array}\right.$$Here $y=\frac{q-1}{q}\cdot x+\frac{1}{q}$, $q_{0}=1+{\left(1-2\epsilon \right)}^{2}$, and $\alpha_{0}$ is determined by $1-\alpha_{0}-\frac{\alpha_{0}\phi_{\epsilon }\left(\alpha_{0}\right)}{1-\alpha_{0}}=y$. Note that $\alpha_{0}$ is well-defined, by the fourth claim of Lemma 3. The fact that $\kappa_{2,q}$ is decreasing in *x* follows from combining the third and the fourth claims of Lemma 3. In fact, $\kappa_{2,q}$ is a constant-$\left(1+{\left(1-2\epsilon \right)}^{2}\right)$ function for $0\leq x\leq \frac{q-q_{0}}{\left(q-1\right)q_{0}}$, and it is strictly decreasing for larger *x*.$q<q_{0}$. In this case *y* is always greater than $\frac{1}{q_{0}}$ and we have that
$$\kappa_{2,q}\left(x,\epsilon \right)\;=\;\left\{\begin{array}{ccc} -\frac{x}{\phi_{\epsilon }\left(1-x\right)} & if & -\frac{x}{\phi_{\epsilon }\left(1-x\right)}\geq \;q\\ \frac{\alpha_{0}-1}{\phi_{\epsilon }\left(\alpha_{0}\right)} & if & -\frac{x}{\phi_{\epsilon }\left(1-x\right)}\leq q \end{array}\right.$$It suffices to show that $\kappa_{2,q}$ is decreasing on both relevant subintervals of [0,1], and this again follows from the third and the fourth claims of Lemma 3. In this case $\kappa_{2,q}$ is strictly decreasing on [0,1].

This completes the proof of Theorem 1. □

## 4. Remaining Proofs

**Proof** **of** **Lemma** **3.**The strict concavity of $\phi_{\epsilon }$ and the bounds on its derivative were shown in [9], Lemma 2.13 (note that $\phi_{\epsilon }\left(x\right)=\frac{1}{2}\tilde{\phi }\left(x,2\epsilon \left(1-\epsilon \right)\right)$ in terms of [9]). The value of $\phi_{\epsilon }$ at the endpoints of the interval [0,1] are directly computable.We pass to the third claim of the lemma. Taking the derivative and rearranging, it suffices to prove that for any α∈(0,1) holds $\phi_{\epsilon }\left(\alpha \right)>\left(\alpha -1\right)\phi_{\epsilon }^{\prime }\left(\alpha \right)$. This follows immediately from the strict concavity of $\phi_{\epsilon }$ and the fact that $\phi_{\epsilon }\left(1\right)=0$.We pass to the last claim of the lemma. Taking the derivative and rearranging, it suffices to prove that for any α∈(0,1) holds
$$\left(1-\alpha \right)\left(\alpha \phi_{\epsilon }^{\prime }\left(\alpha \right)+\left(1-\alpha \right)\right)>-\phi_{\epsilon }\left(\alpha \right).$$Since $\left(1-\alpha \right)\cdot \phi_{\epsilon }^{\prime }\left(\alpha \right)>-\phi_{\epsilon }\left(\alpha \right)$, it suffices to show that $\alpha \phi_{\epsilon }^{\prime }\left(\alpha \right)+\left(1-\alpha \right)\geq \phi_{\epsilon }^{\prime }\left(\alpha \right)$, and this follows from the first two claims of the lemma. The values of the function *g* at the endpoints are directly computable. □

**Proof** **of** **Lemma** **1.**We start with a technical lemma which deals with the behavior of the function $\phi_{\epsilon }\left(x\right)$ in the vicinity of ϵ=0. We write ϵ∼0 as a shorthand for “ϵ close to 0”. We again use the fact that $\phi_{\epsilon }\left(x\right)=\Phi \left(x,2\epsilon \left(1-\epsilon \right)\right)=\frac{1}{2}\tilde{\phi }\left(x,2\epsilon \left(1-\epsilon \right)\right)$, where the function $\tilde{\phi }$ was defined and studied in [9]. In the calculations below ϕ(x,ϵ) is written instead of $\phi_{\epsilon }\left(x\right)$, for notational convenience.

**Lemma** **9.**
*Let 0<t<1. Then*

*1.* 

$$\phi \left(t,0\right)\;=\;\frac{t-1}{2}\;and\;for\;\epsilon ∼0\;holds\;|\phi \left(t,\epsilon \right)-\frac{t-1}{2}|\leq O\left(\epsilon \right).$$

*2.* 

$$\frac{\partial \phi }{\partial \epsilon }\left(t,0\right)\;=\;\frac{2\sqrt{H^{-1}\left(t\right)\left(1-H^{-1}\left(t\right)\right)}-1}{\ln(2)}\;and\;for\;\epsilon ∼0\;holds\;$$


$$|\frac{\partial \phi }{\partial \epsilon }\left(t,\epsilon \right)-\frac{2\sqrt{H^{-1}\left(t\right)\left(1-H^{-1}\left(t\right)\right)}-1}{\ln(2)}|\leq O\left(\epsilon \right).$$

*3.* 

$$\frac{\partial \phi }{\partial t}\left(t,0\right)\;=\;\frac{1}{2}\;and\;for\;\epsilon ∼0\;holds\;|\frac{\partial \phi }{\partial t}\left(t,\epsilon \right)-\frac{1}{2}|\leq O\left(\epsilon \right).$$




**Proof** **of** **Lemma** **9.***Notation*. Here and below we write a±ϵ as a shorthand for the interval [a−ϵ,a +ϵ].Recall that
$$\tilde{\phi }\left(t,\epsilon \right)\;=\;t-1+\sigma H\left(\frac{z}{\sigma }\right)+\left(1-\sigma \right)H\left(\frac{z}{1-\sigma }\right)+2z\log_{2}\left(\epsilon \right)+\left(1-2z\right)\log_{2}\left(1-\epsilon \right),$$
where $\sigma =H^{-1}\left(t\right)$ and $z=z\left(t,\epsilon \right)=\frac{-\epsilon^{2}+\epsilon \sqrt{\epsilon^{2}+4\left(1-2\epsilon \right)\sigma \left(1-\sigma \right)}}{2(1-2\epsilon )}$.The fact that $\phi \left(t,0\right)=\frac{1}{2}\tilde{\phi }\left(t,0\right)=\frac{t-1}{2}$ is verified by inspection, observing that z(t,0)=0 for any *t*. Note also that, by assumption, σ>0, and hence $z\left(t,\epsilon \right)\in \sqrt{\sigma (1-\sigma )}\cdot \epsilon \pm O\left(\epsilon^{2}\right)$ for a sufficiently small ϵ.Using (as in the proof of Lemma 2.13 in [9]) the fact that for ϵ>0 holds $\frac{\left(\sigma -z)(1-\sigma -z\right)}{z^{2}}=\frac{{\left(1-\epsilon \right)}^{2}}{\epsilon^{2}}$, and writing δ=2ϵ(1−ϵ), we have that
$$\frac{\partial \phi (t,\epsilon )}{\partial \epsilon }\;=\;\frac{1}{2}\cdot \frac{\partial \tilde{\phi }\left(t,\delta \right)}{\partial \epsilon }\;=\;\frac{1-2\epsilon }{\ln(2)}\cdot \frac{2z-\delta }{\delta (1-\delta )}.$$Hence for ϵ∼0 we have $\frac{\partial \phi (t,\epsilon )}{\partial \epsilon }\in \frac{2\sqrt{\sigma (1-\sigma )}-1}{\ln(2)}\pm O\left(\delta \right)$, or equivalently $\frac{\partial \phi (t,\epsilon )}{\partial \epsilon }\in \frac{2\sqrt{\sigma (1-\sigma )}-1}{\ln(2)}\pm O\left(\epsilon \right)$.In particular,
$${\frac{\partial \phi (t,\epsilon )}{\partial \epsilon }}_{|\epsilon =0}\;=\;\lim_{\epsilon \to 0}\frac{\partial \phi (t,\epsilon )}{\partial \epsilon }\;=\;\frac{2\sqrt{\sigma (1-\sigma )}-1}{\ln(2)}\;=\;\frac{2\sqrt{H^{-1}\left(t\right)\left(1-H^{-1}\left(t\right)\right)}-1}{\ln(2)}.$$
This proves both the first and the second claims of the lemma.We pass to the third claim of the lemma. As shown in the proof of Lemma 2.13 in KS2 we have $\frac{\partial \tilde{\phi }}{\partial t}\left(t,\epsilon \right)=\frac{\ln\left(\frac{1-\sigma -z}{\sigma -z}\right)}{\ln\left(\frac{1-\sigma }{\sigma }\right)}$. Hence
$$\frac{\partial \phi (t,\epsilon )}{\partial t}\;=\;\frac{1}{2}\cdot \frac{\partial \tilde{\phi }\left(t,\delta \right)}{\partial t}\;=\;\frac{1}{2}\cdot \frac{\ln\left(\frac{1-\sigma -z}{\sigma -z}\right)}{\ln\left(\frac{1-\sigma }{\sigma }\right)},$$
where z=z(t,δ). Recall for any 0<t<1 we have z(t,0)=0 and in addition for δ∼0 we have $z\left(t,\delta \right)\in \sqrt{\sigma (1-\sigma )}\cdot \delta \pm O\left(\delta^{2}\right)$. The third claim of the lemma now follows by inspection. This completes the proof of Lemma 9. □

We proceed with the proof of Lemma 1. First, consider the definition of $\kappa =\kappa_{2,2}$. For ϵ sufficiently close to zero, we have that $\frac{x+1}{2}>\frac{1}{q_{0}}$ (recall that $q_{0}=1+{\left(1-2\epsilon \right)}^{2}$) and hence $\kappa =\frac{\alpha -1}{\phi (\alpha ,\epsilon )}$, where α=α(ϵ) is determined by $1-\alpha +\frac{\alpha \phi (\alpha ,\epsilon )}{\alpha -1}=\frac{x+1}{2}$. Taking the derivative w.r.t. ϵ in the definition of α and rearranging gives
$$\alpha^{\prime }\left(\epsilon \right)\;=\;-\frac{\alpha \left(\alpha -1\right)\frac{\partial \phi }{\partial \epsilon }\left(\alpha ,\epsilon \right)}{\alpha \left(\alpha -1\right)\frac{\partial \phi }{\partial \alpha }\left(\alpha ,\epsilon \right)-\phi \left(\alpha ,\epsilon \right)-{\left(\alpha -1\right)}^{2}}.$$

Using the first claim of Lemma 9, it is easy to see that α(0)=1−x. Hence, using all claims of Lemma 9, we have that
$$\alpha^{\prime }\left(0\right)\;=\;-\frac{2}{\ln2}\cdot \frac{\alpha \left(0\right)\left(2\sqrt{H^{-1}\left(\alpha (0)\right)\left(1-H^{-1}\left(\alpha (0)\right)\right)}-1\right)}{1-\alpha (0)}\;=\;$$
$$-\frac{2}{\ln2}\cdot \frac{\left(1-x\right)\left(2\sqrt{H^{-1}\left(1-x\right)\left(1-H^{-1}\left(1-x\right)\right)}-1\right)}{x}$$

Next, we compute κ and $\kappa^{\prime }$ at 0. Note that by the definition of κ, we have $1-\alpha +\frac{\alpha }{\kappa }=\frac{x+1}{2}$. Hence, $\kappa =\frac{\alpha }{\frac{x+1}{2}+\alpha -1}$ and $\kappa^{\prime }=\frac{\kappa \left(1-\kappa \right)\alpha^{\prime }}{\alpha }$. In particular, κ(0)=2 and
$$\kappa^{\prime }\left(0\right)\;=\;\;=\;-\frac{2\alpha^{\prime }\left(0\right)}{\alpha (0)}\;=\;\frac{4}{\ln2}\cdot \frac{\left(2\sqrt{H^{-1}\left(1-x\right)\left(1-H^{-1}\left(1-x\right)\right)}-1\right)}{x},$$
proving the first claim of the proposition.

Let now ϵ∼0. We start with estimating α(ϵ) and κ(ϵ). From the identity $1-\alpha +\frac{\alpha \phi (\alpha ,\epsilon )}{\alpha -1}=\frac{x+1}{2}$, using the monotonicity of the LHS in α (by Lemma 2.3) and Lemma 9, it is easy to see that α(ϵ)∈1−x±O(ϵ). From this, and from the identity $1-\alpha +\frac{\alpha }{\kappa }=\frac{x+1}{2}$, we get $\kappa \left(\epsilon \right)=\frac{\alpha (\epsilon )}{\frac{x+1}{2}+\alpha \left(\epsilon \right)-1}\in 2\pm O\left(\epsilon \right)$.

Proceeding in a similar vein, using the above expression for $\alpha^{\prime }$, we get that
$$\alpha^{\prime }\left(\epsilon \right)\;\in \;\frac{2}{\ln(2)}\cdot \frac{1-x}{x}\cdot \left(1-2\sqrt{H^{-1}\left(1-x\right)\left(1-H^{-1}\left(1-x\right)\right)}\right)\pm O\left(\epsilon \right),$$
and
$$\kappa^{\prime }\left(\epsilon \right)\;=\;\frac{\kappa \left(\epsilon \right)\left(1-\kappa \left(\epsilon \right)\right)\alpha^{\prime }\left(\epsilon \right)}{\alpha (\epsilon )}\in \;=\;-\frac{2\alpha^{\prime }\left(\epsilon \right)}{\alpha (\epsilon )}\;\subseteq \;$$
$$\frac{4}{\ln2}\cdot \frac{\left(2\sqrt{H^{-1}\left(1-x\right)\left(1-H^{-1}\left(1-x\right)\right)}-1\right)}{x}\pm O\left(\epsilon \right),$$
completing the proof of the lemma. □

**Proof** **of** **Corollary** **2.**Let q=2 and $\kappa =\kappa_{2,2}$ (see the second claim of Corollary 1 for a more explicit statement of Theorem 1 in this case). Viewing both sides of (Equation 5) as functions of ϵ, and writing L(ϵ) for the LHS and R(ϵ) for the RHS, we have $L\left(0\right)=R\left(0\right)={∥f∥}_{2}$, and L(ϵ)≤R(ϵ) for $0\leq \epsilon \leq \frac{1}{2}$. It is easy to see that both *L* and *R* are differentiable, and we may deduce that $L^{\prime }\left(0\right)\leq R^{\prime }\left(0\right)$. Computing the derivatives (see, e.g., [3]) gives
$$L^{\prime }\left(0\right)\;=\;-\frac{1}{2}\cdot \frac{E(f,f)}{{∥f∥}_{2}}\;\mathrm{and}\;R^{\prime }\left(0\right)\;=\;\frac{\ln\left(2\right)\kappa^{\prime }\left(0\right)}{4}\cdot \frac{Ent\left(f^{2}\right)}{{∥f∥}_{2}},$$
where we write $\kappa^{\prime }\left(0\right)$ for ${\frac{\partial \kappa }{\partial \epsilon }}_{|\epsilon =0}$. Hence $L^{\prime }\left(0\right)\leq R^{\prime }\left(0\right)$ is equivalent to
(7)$$E\left(f,f\right)\;\geq \;-\frac{\ln\left(2\right)\kappa^{\prime }\left(0\right)}{2}\cdot Ent\left(f^{2}\right).$$The claim of the corollary now follows from the first claim of Lemma 1. It only remains to add that the fact that ℓ(·) a convex and increasing function on [0,1], taking [0,1] onto $\left[2\ln2,2\right]$ was proved in [20]. □

**Proof** **of** **Corollary** **3.**Let us point out that our argument follows along the same lines as the proof of the same result in [10]. We do believe that the argument here is worth presenting in full, since it seems to be somewhat more explicit and easier to parse.We use the simple fact (see, e.g., [4]) that for any $0\leq \epsilon \leq \frac{1}{2}$ and for any $\alpha \in {\left\{0,1\right\}}^{n}$ holds $\hat{f_{\epsilon }}\left(\alpha \right)={\left(1-2\epsilon \right)}^{\left|\alpha \right|}\hat{f}\left(\alpha \right)$. Hence, using Parseval’s identity in the first step below, we have
$$∥f_{\epsilon }{∥}_{2}^{2}\;=\;\sum_{\alpha \in {\left\{0,1\right\}}^{n}}{\left(1-2\epsilon \right)}^{2|\alpha |}{\hat{f}}^{2}\left(\alpha \right)\;\geq \;{\left(1-2\epsilon \right)}^{2\mu n}\cdot \sum_{|\alpha |\leq \mu n}{\hat{f}}^{2}\left(\alpha \right).$$
Since this holds for any $0\leq \epsilon \leq \frac{1}{2}$, we deduce that
$$\sum_{|\alpha |\leq \mu n}{\hat{f}}^{2}\left(\alpha \right)\;\leq \;\min_{0\leq \epsilon \leq \frac{1}{2}}\frac{∥f_{\epsilon }{∥}_{2}^{2}}{{\left(1-2\epsilon \right)}^{2\mu n}}\;\leq \;\min_{0\leq \epsilon \leq \frac{1}{2}}\frac{{∥f∥}_{\kappa }^{2}}{{\left(1-2\epsilon \right)}^{2\mu n}},$$
where we have used Theorem 1 with q=2 in the second step, and $\kappa =\kappa \left(\epsilon \right)=\kappa_{2,2}\left(\frac{Ent_{2}\left(\frac{f}{{∥f∥}_{1}}\right)}{n},\epsilon \right)$.Let $F\left(\epsilon \right)=\frac{1}{n}\log_{2}\left(\frac{{∥f∥}_{\kappa }^{2}}{{\left(1-2\epsilon \right)}^{2\mu n}}\right)=\frac{1}{n}\log_{2}\left({∥f∥}_{\kappa }^{2}\right)-2\mu \log_{2}\left(1-2\epsilon \right)$. Since κ(0)=2, we have $F\left(0\right)=\frac{1}{n}\log_{2}\left({∥f∥}_{2}^{2}\right)$. Hence the claim of the corollary is equivalent to the claim that $\min_{0\leq \epsilon \leq \frac{1}{2}}F\left(\epsilon \right)$ is negative and bounded away from F(0) by some absolute constant. To show this, it suffices to show that $F^{\prime }\left(\epsilon \right)$ is negative and bounded away from 0 by an absolute constant for ϵ in a constant length interval $\left[0,\epsilon_{0}\right]$.Recall that for any nonnegative non-zero function *g* on ${\left\{0,1\right\}}^{n}$ holds $\frac{Ent\left(g^{2}\right)}{Eg^{2}}\geq \log_{2}\left(\frac{Eg^{2}}{E^{2}g}\right)=Ent_{2}\left(\frac{g}{{∥g∥}_{1}}\right)$ (see, e.g., [10]). Recall also that $\frac{\partial }{\partial \epsilon }\log_{2}\left({∥f∥}_{\kappa (\epsilon )}\right)=\frac{\kappa^{\prime }}{\kappa^{2}}\cdot \frac{Ent\left({\left|f\right|}^{\kappa }\right)}{{∥f∥}_{\kappa }^{\kappa }}$.Hence, recalling that, by Lemma 1, $\kappa^{\prime }<0$ in the vicinity of 0, we have
$$F^{\prime }\left(\epsilon \right)\;=\;2\frac{\kappa^{\prime }}{\kappa^{2}}\cdot \frac{1}{n}\frac{Ent\left({\left|f\right|}^{\kappa }\right)}{{∥f∥}_{\kappa }^{\kappa }}+\frac{4}{\ln(2)}\cdot \frac{\mu }{1-2\epsilon }\;\leq \;2\frac{\kappa^{\prime }}{\kappa^{2}}\cdot \frac{1}{n}\log_{2}\left(\frac{E\left({\left|f\right|}^{\kappa }\right)}{E^{2}{\left|f\right|}^{\kappa /2}}\right)+\frac{4}{\ln(2)}\cdot \frac{\mu }{1-2\epsilon }.$$Let $x=\frac{Ent_{2}\left(\frac{f}{{∥f∥}_{1}}\right)}{n}=1-H\left(\rho \right)$. Recalling again that κ(0)=2 and applying the first claim of Lemma 1 we get
$$F^{\prime }\left(0\right)\;\leq \;\frac{\kappa^{\prime }\left(0\right)}{2}\cdot x+\frac{4\mu }{\ln(2)}\;=\;\frac{4}{\ln(2)}\cdot \left(\mu -\left(\frac{1}{2}-\sqrt{\rho (1-\rho )}\right)\right)<0.$$It now suffices to show that for sufficiently small ϵ we have $F^{\prime }\left(\epsilon \right)\leq F^{\prime }\left(0\right)+O\left(\epsilon \right)$. Taking the second claim of Lemma 1 into account, it is enough to show that $\frac{1}{n}\log_{2}\left(\frac{E\left({\left|f\right|}^{\kappa }\right)}{E^{2}{\left|f\right|}^{\kappa /2}}\right)\geq x-O\left(\epsilon \right)$. Let $G\left(\epsilon \right)=\frac{1}{n}\log_{2}\left(\frac{E\left({\left|f\right|}^{\kappa }\right)}{E^{2}{\left|f\right|}^{\kappa /2}}\right)$. Then G(0)=x and it suffices to show that $|G^{\prime }|$ is bounded by an absolute constant. A simple calculation gives that
$$G^{\prime }\;=\;\frac{\kappa^{\prime }}{\kappa }\cdot \left(\frac{1}{n}\frac{Ent\left({\left|f\right|}^{\kappa }\right)}{{E|f|}^{\kappa }}-\frac{2}{n}\frac{Ent\left({\left|f\right|}^{\kappa /2}\right)}{{E|f|}^{\kappa /2}}+G\right).$$The RHS in the last expression is bounded by a constant, since for any nonnegative non-zero function *g* on ${\left\{0,1\right\}}^{n}$ both $\frac{Ent(g)}{Eg}$ and $\log_{2}\left(\frac{Eg^{2}}{E^{2}\left(g\right)}\right)$ are bounded by *n*. □

**Proof** **of** **Corollary** **4.** 
**The first claim of the corollary**
Let $D\subseteq {\left\{0,1\right\}}^{n}$, $\left|D\right|=2^{H(\rho )n}$. Let $M_{D}$ be the adjacency matrix of the subgraph of the discrete cube induced by the vertices of *D*. Let λ(D) be the maximal eigenvalue of $M_{D}$. Let *f* be a maximal eigenvector of $M_{D}$. We view *f* as a function on *D* and extend it to a function on ${\left\{0,1\right\}}^{n}$ by defining it to be zero outside *D*. Let *A* be the adjacency matrix of ${\left\{0,1\right\}}^{n}$. Then $\lambda \left(D\right)=\frac{\langle f,M_{D}f\rangle }{\langle f,f\rangle }=\frac{\langle f,Af\rangle }{\langle f,f\rangle }$. Note also that since *f* is supported on *D* we have
$$E^{2}\left|f\right|\;=\;{\left(\langle f,\mathrm{sign}\left(f\right)\cdot 1_{D}\rangle \right)}^{2}\;\leq \;Ef^{2}\cdot E{\left(\mathrm{sign}\left(f\right)\cdot 1_{D}\right)}^{2}\;=\;Ef^{2}\cdot \frac{\left|D\right|}{2^{n}}\;=\;Ef^{2}\cdot 2^{(H(\rho )-1)n}.$$
It follows that $\frac{Ent_{2}\left(\frac{f}{{∥f∥}_{1}}\right)}{n}\geq 1-H\left(\rho \right)$.Next, it is easy to check that for any function *g* on ${\left\{0,1\right\}}^{n}$ holds $E\left(g,g\right)=2\langle g,(nI-A)g\rangle$, where *I* is the $2^{n}\times 2^{n}$ identity matrix. Hence, using Corollary 2 and the fact that $\frac{Ent\left(f^{2}\right)}{Ef^{2}}\geq \log_{2}\left(\frac{Ef^{2}}{E^{2}\left|f\right|}\right)=Ent_{2}\left(\frac{f}{{∥f∥}_{1}}\right)$, we have, writing *x* for $\frac{1}{n}Ent_{2}\left(\frac{f}{{∥f∥}_{1}}\right)$,
$$\lambda \left(D\right)\;=\;\frac{\langle f,Af\rangle }{\langle f,f\rangle }\;=\;n-\frac{1}{2}\frac{E(f,f)}{\langle f,f\rangle }\;\leq \;n-\frac{1}{2}\frac{\ell \left(x\right)\cdot Ent\left(f^{2}\right)}{Ef^{2}}\;\leq \;$$
$$n-\frac{n}{2}x\ell \left(x\right)\;\leq \;n\left(1-\frac{1}{2}(1-H\left(\rho \right)\ell \left(1-H(\rho )\right)\right)\;=\;2\sqrt{\rho (1-\rho )}\cdot n.$$This is almost tight if we set r=⌈ρn⌉ and take $D=\left\{x\in {\left\{0,1\right\}}^{n}:\;\left|x\right|\leq r\right\}$ to be the Hamming ball of radius *r* around 0. In fact, recall that $\left|D\right|\approx 2^{H(\rho )n}$ (see, e.g., [19]) and, as shown in [19], $\lambda \left(D\right)\geq 2\sqrt{\rho (1-\rho )}\cdot n-o\left(n\right)$. □


**The second claim of the corollary**


Let $0<\delta <\frac{1}{2}$. Let d=⌊δn⌋, and let *f* be a feasible solution of the dual linear program of [15] with parameters *n* and *d*. Then, as observed by [29] *f* can be viewed as a function on ${\left\{0,1\right\}}^{n}$ with the following properties:*f* is symmetric, that is f(x) depends only on |x|.f(x)≤0 for |x|≥d.$\hat{f}\geq 0$ and $\hat{f}\left(0\right)=1$.$f\left(0\right)\leq 2^{R_{LP}\left(\delta \right)\cdot n+o\left(n\right)}$.

To prove the claim, we will show that any function *f* with the first three of these properties satisfies $\frac{1}{n}\log_{2}\left(f\left(0\right)\right)\geq \frac{1-H\left(\delta \right)+H\left(\frac{1}{2}-\sqrt{\delta (1-\delta )}\right)}{2}-o_{n}\left(1\right)$.

*Notation*: We write ${∥g∥}_{q,F}$ for ${\left(\sum_{\alpha \in {\left\{0,1\right\}}^{n}}{\left|g\left(\alpha \right)\right|}^{q}\right)}^{1/q}$. Note that Parseval’s identity states ${∥f∥}_{2}={∥\hat{f}∥}_{2,F}$. We write ≈, ≲, and ≳ to denote equality or inequality which hold up to lower order terms. To give an example, recall that for $0<\rho \leq \frac{1}{2}$ the cardinalities of the Hamming ball $\left\{x\in {\left\{0,1\right\}}^{n}:\;\left|x\right|\leq r\right\}$ and the Hamming sphere $\left\{x\in {\left\{0,1\right\}}^{n}:\;\left|x\right|=r\right\}$ are $2^{H(\rho )n}$, up to lower order terms. We write this as $\frac{1}{n}\log_{2}\left(|\left\{x\in {\left\{0,1\right\}}^{n}:\;\left|x\right|\leq r\right\}|\right)\approx H\left(\rho \right)$.

We start with some preliminary observations. First, we need some simple and well-known facts from Fourier analysis on ${\left\{0,1\right\}}^{n}$. If *f* is symmetric, then so is $\hat{f}$. Next, $\hat{f}\left(0\right)=Ef\leq {∥f∥}_{1}$. Furthermore, finally, using the fact that in our case $\hat{f}\geq 0$, $f\left(0\right)=\sum_{\alpha \in {\left\{0,1\right\}}^{n}}\hat{f}\left(\alpha \right)={∥\hat{f}∥}_{1,F}$.

Next, we claim that if *f* is symmetric and if, for some 0≤i≤n holds 12nni|f(i)|≥Ω1n·∥f∥1 then ∥f∥2∥f∥1≥Ω1n·2nni. In fact, we will have
∥f∥22≥12nnif2(i)≥Ω1n2·12nni2nni∥f∥12=Ω1n2·2nni∥f∥12.

Similarly, if for some 0≤j≤n holds njf^2(j)≥Ω1n·∥f^∥2,F2 then ∥f^∥1,F∥f^∥2,F≥Ω1n·nj.

Finally, we need a slight extension of Corollary 3. As stated, it shows that if *f* has a large second entropy, then $\hat{f}$ cannot attain its $\ell_{2}$ norm in a Hamming ball of small radius *around 0*. We claim, as was also observed in [10], that this holds more generally for Hamming balls with arbitrary centers in ${\left\{0,1\right\}}^{n}$. To see that, let $z\in {\left\{0,1\right\}}^{n}$, and define $g=f\cdot W_{z}$, where $W_{z}$ is the corresponding Walsh-Fourier character. It is easy to see that for any $y\in {\left\{0,1\right\}}^{n}$ holds $\hat{g}\left(y\right)=\hat{f}\left(y+z\right)$, and hence *g* has the same first and second norms as *f*. Moreover, writing B(z,r) for the Hamming ball of radius *r* around *z*, we have $\sum_{\alpha \in B(z,r)}{\hat{f}}^{2}\left(\alpha \right)=\sum_{\beta \in B(0,r)}{\hat{g}}^{2}\left(\beta \right)$.

We pass to the proof of the claim. Note that since f(x)≤0 for |x|≥d and since Ef≥0, there exists 0≤i≤d−1 such that 12nni|f(i)|≥Ω1n·∥f∥1. Hence
$$\frac{1}{n}Ent_{2}\left(\frac{f}{{∥f∥}_{1}}\right)\;=\;\frac{1}{n}\log_{2}\left(\frac{{∥f∥}_{2}^{2}}{{∥f∥}_{1}^{2}}\right)\;≳\;1-H\left(\frac{i}{n}\right)\;\geq \;1-H\left(\delta \right).$$

By Corollary 3 this means that $\hat{f}$ cannot attain its $\ell_{2}$ norms inside Hamming balls or radii much smaller than $r\left(\delta \right):=\left(\frac{1}{2}-\sqrt{\delta (1-\delta )}\right)\cdot n$ around the all-0 and all-1 vectors. Hence there exists r(δ)−o(n)≤j≤r(δ)+o(n) such that njf^2(j)≥Ω1n·∥f^∥2,F2. It follows that
$$\frac{1}{n}\log_{2}\left(\frac{∥\hat{f}{∥}_{1,F}}{∥\hat{f}{∥}_{2,F}}\right)\;≳\;\frac{H\left(\frac{j}{n}\right)}{2}\;≳\;\frac{H\left(\frac{1}{2}-\sqrt{\delta (1-\delta )}\right)}{2}.$$

We can now complete the proof of the second claim of the corollary. We have
$$0\;=\;\frac{1}{n}\log_{2}\left(\hat{f}\left(0\right)\right)\;\leq \;\frac{1}{n}\log_{2}\left({∥f∥}_{1}\right)\;≲\;\frac{1}{n}\log_{2}\left({∥f∥}_{2}\right)-\frac{1-H(\delta )}{2}\;=\;$$
$$\frac{1}{n}\log_{2}\left(∥\hat{f}{∥}_{2,F}\right)-\frac{1-H(\delta )}{2}\;≲\;\frac{1}{n}\log_{2}\left(∥\hat{f}{∥}_{1,F}\right)-\frac{1-H\left(\delta \right)+H\left(\frac{1}{2}-\sqrt{\delta (1-\delta )}\right)}{2}\;=\;$$
$$\frac{1}{n}\log_{2}\left(f(0)\right)-\frac{1-H\left(\delta \right)+H\left(\frac{1}{2}-\sqrt{\delta (1-\delta )}\right)}{2}.$$ □

**Proof** **of** **Corollary** **5.**Let 0≤s≤n/2 and let *f* be a polynomial of degree *s* on ${\left\{0,1\right\}}^{n}$. We need two simple and well-known facts from Fourier analysis on ${\left\{0,1\right\}}^{n}$. First, that the Fourier expansion of *f* is supported on characters of weight at most *s*; and second, that for any function *g* on ${\left\{0,1\right\}}^{n}$ holds $E\left(g,g\right)=4\sum_{\alpha \in {\left\{0,1\right\}}^{n}}\left|\alpha \right|{\hat{g}}^{2}\left(\alpha \right)$. Combining these two facts implies that
$$E\left(f,f\right)\;=\;4\sum_{\alpha \in {\left\{0,1\right\}}^{n}}\left|\alpha \right|{\hat{f}}^{2}\left(\alpha \right)\;=\;4\sum_{|\alpha |\leq s}\left|\alpha \right|{\hat{f}}^{2}\left(\alpha \right)\leq 4s\cdot \sum_{|\alpha |\leq s}{\hat{f}}^{2}\left(\alpha \right)\;=\;4s\cdot Ef^{2},$$
where in the last step we used Parseval’s identity.Write σ for s/n and *x* for $\frac{1}{n}Ent_{2}\left(\frac{f}{{∥f∥}_{1}}\right)$. We have, using Corollary 2,
$$4\sigma n\;=\;4s\;\geq \;\frac{E(f,f)}{Ef^{2}}\;\geq \;\ell \left(x\right)\cdot \frac{Ent\left(f^{2}\right)}{Ef^{2}}\;\geq \;nx\ell \left(x\right)\;=$$
$$n\cdot \left(2-4\sqrt{H^{-1}\left(1-x\right)\left(1-H^{-1}\left(1-x\right)\right)}\right).$$Rearranging and simplifying, this is equivalent to
$$\frac{1}{n}\log_{2}\left(\frac{{∥f∥}_{2}}{{∥f∥}_{1}}\right)\;=\;\frac{x}{2}\;\leq \;\frac{1-H\left(\frac{1}{2}-\sqrt{\sigma (1-\sigma )}\right)}{2},$$
completing the proof. □

**Proof** **of** **Corollary** **1.**We start with the first claim of the corollary. First consider the case $\epsilon =\frac{1}{2}$. It is easy to see that $\phi_{\frac{1}{2}}\left(x\right)=x-1$ (note that in the definition of Φ(x,ϵ) we have $y\left(x,\frac{1}{2}\right)=\lim_{\epsilon \to \frac{1}{2}}y\left(x,\epsilon \right)=H^{-1}\left(x\right)\left(1-H^{-1}\left(x\right)\right)$) and hence in this case the value of κ given by the claim is 1 (as it should be).Assume now $\epsilon <\frac{1}{2}$. This implies that $q_{0}=1+{\left(1-2\epsilon \right)}^{2}>1$. By the first claim of Lemma 3, this means that for any 0≤x≤1 we have $\frac{-x}{\phi_{\epsilon }\left(1-x\right)}\geq -\frac{1}{\phi_{\epsilon }\left(0\right)}=\frac{2}{\log_{2}\left(\frac{4}{q_{0}}\right)}>1$. Hence, it is easy to see that for *q* sufficiently close to 1 the first and the third clauses in the definition of $\kappa_{2,q}$ in Definition 1 do not apply, and we have $\kappa_{2,q}\left(x,\epsilon \right)=\frac{-x}{\phi_{\epsilon }\left(1-x\right)}$. Theorem 1 then gives
$$∥f_{\epsilon }{∥}_{2}\;\leq \;{∥f∥}_{\kappa },\;with\;\kappa \;=\;-\frac{\frac{Ent_{q}\left(\frac{f}{{∥f∥}_{1}}\right)}{n}}{\phi_{\epsilon }\left(1-\frac{Ent_{q}\left(\frac{f}{{∥f∥}_{1}}\right)}{n}\right)}.$$Taking q→1 and recalling that $Ent_{q}\left(\cdot \right)\to_{q\to 1}Ent\left(\cdot \right)$ completes the proof of the claim.We pass to the second claim of the corollary. First consider the case ϵ=0. Note that in this case $q_{0}=2$. Furthermore, by the first claim of Lemma 9, $\phi_{0}\left(x\right)=\frac{x-1}{2}$, and hence the value of κ given by the claim is 2 (as expected).Assume now ϵ>0. This implies that $q_{0}<2$, and hence, by the third claim of Lemma 3, for any 0≤x≤1 we have $\frac{-x}{\phi_{\epsilon }\left(1-x\right)}\leq q_{0}<2=q$. Hence the second clause in the definition of $\kappa_{2,q}$ in Definition 1 does not apply. The remaining two clauses give the claim, as stated. □

### Proofs of Comments to Theorem 1

Some of the claims in these comments require a proof. These claims are restated and proved in the following lemma.

**Lemma** **10.**

*If q≥2 then for any $0<\epsilon <\frac{1}{2}$ the function $\kappa_{2,q}\left(x,\epsilon \right)$ starts as a constant-$\left(1+{\left(1-2\epsilon \right)}^{2}\right)$ function up to some x=x(q,ϵ)>0, and becomes strictly decreasing after that. For 1<q<2 there is a value $0<\epsilon \left(q\right)<\frac{1}{2}$, such that for all ϵ≤ϵ(q) the function $\kappa_{2,q}\left(x,\epsilon \right)$ is strictly decreasing (in which case we say that x(q,ϵ)=0). However, x(q,ϵ)>0 for all ϵ>ϵ(q). The function ϵ(q) decreases with q (in particular, ϵ(q)=0 for g≥2). The function x(q,ϵ) increases both in q and in ϵ.*

*The function $\kappa_{2,1}\left(x,\epsilon \right)=-\frac{x}{\phi_{\epsilon }\left(1-x\right)}$ is strictly decreasing in its first argument for any $0<\epsilon <\frac{1}{2}$. It satisfies $\kappa_{2,1}\left(0,\epsilon \right)=\lim_{x\to 0}\kappa_{2,1}\left(x,\epsilon \right)=1+{\left(1-2\epsilon \right)}^{2}$, for all $0\leq \epsilon \leq \frac{1}{2}$.*

*Let f be a non-constant function on ${\left\{0,1\right\}}^{n}$. Let $0<\epsilon <\frac{1}{2}$. Let $F\left(q\right)=F_{f,\epsilon }\left(q\right)=\kappa_{2,q}\left(Ent_{q}\left(\frac{f}{{∥f∥}_{1}}\right)/n,\epsilon \right)$. There is a unique value $1<q\left(f,\epsilon \right)\leq 1+{\left(1-2\epsilon \right)}^{2}$ of q for which F(q)=q. Moreover, $q\left(f,\epsilon \right)=\min_{q\geq 1}F\left(q\right)$. Furthermore, $\lim_{\epsilon \to 0}q\left(f,\epsilon \right)=2$ for any f.*



**Proof.** The first claim of the lemma follows from the properties of $\kappa_{2,q}$ as shown in the proof of Theorem 1. In particular, it is easy to see that for q≤2 we have $\epsilon \left(q\right)=\frac{1-\sqrt{q-1}}{2}$ and for ϵ≥ϵ(q) we have $x\left(q,\epsilon \right)=\frac{q-\left(1+{\left(1-2\epsilon \right)}^{2}\right)}{\left(1+{\left(1-2\epsilon \right)}^{2}\right)\cdot \left(q-1\right)}$. The claim that ϵ(q) decreases with *q* and that x(q,ϵ) increases in both *q* and ϵ follows by direct verification.The second claim of the lemma follows immediately from the third claim of Lemma 3.We pass to the third claim of the lemma. Note that the function $x\left(q\right)=Ent_{q}\left(\frac{f}{{∥f∥}_{1}}\right)/n$ is positive and strictly increasing in *q*. We need the following auxiliary claim.

**Lemma** **11.**
*The function $y\left(q\right)=\frac{q-1}{q}\cdot x\left(q\right)+\frac{1}{q}$ is strictly decreasing in q.*


**Proof** **of** **Lemma** **11.**Assume w.l.o.g. that f≥0 and that Ef=1. Let $P=\frac{f}{2^{n}}$ be a distribution on ${\left\{0,1\right\}}^{n}$. A simple calculation gives that
$$y\left(q\right)\;=\;1+\frac{1}{n}\cdot \log_{2}\left({\left(\sum_{a\in {\left\{0,1\right\}}^{n}}P{\left(a\right)}^{q}\right)}^{\frac{1}{q}}\right),$$
which is strictly decreasing in *q*, by Hölder’s inequality. □

We proceed with the proof of of the third claim of Lemma 10. Let $q_{0}=1+{\left(1-2\epsilon \right)}^{2}$. We claim, first, that *F* is strictly increasing on $q_{0}\leq q<\infty$. In fact, for these values of *q* the second clause of Definition 1 does not apply (by the third claim of Lemma 3) and we have
$$\kappa_{2,q}\left(x,\epsilon \right)\;=\;\left\{\begin{array}{ccc} q_{0} & \mathrm{if} & y\leq \frac{1}{q_{0}}\\ \frac{\alpha_{0}-1}{\phi_{\epsilon }\left(\alpha_{0}\right)} & \mathrm{if} & y>\frac{1}{q_{0}} \end{array}\right.,$$
where y=y(q) and $\alpha_{0}$ is determined by $1-\alpha_{0}-\frac{\alpha_{0}\phi_{\epsilon }\left(\alpha_{0}\right)}{1-\alpha_{0}}=y$. The claim now follows by combining Lemma 11, and the third and fourth claims of Lemma 3.

Next, we claim that there exists a unique value $1\leq q=q^{*}\leq q_{0}$ for which $\frac{-x}{\phi_{\epsilon }\left(1-x\right)}=q$ (here x=x(q)). Moreover, *F* decreases for $1\leq q\leq q^{*}$ and increases for $q\geq q^{*}$. Finally, $F\left(q^{*}\right)=q^{*}$. Observe that verifying these claims will essentially complete the proof of the third claim of Lemma 10 (apart from the fact that $\lim_{\epsilon \to 0}q\left(f,\epsilon \right)=2$).

In fact, by the first and third claims of Lemma 3, and the fact that *x* is strictly increasing in *q*, the function $\frac{-x}{\phi_{\epsilon }\left(1-x\right)}$ is strictly decreasing in *q*, taking values between $\frac{2}{\log_{2}\left(\frac{4}{q_{0}}\right)}$ and $q_{0}$. This means that it has a unique intersection $q=q^{*}$ with the function *q* in $\left[1,q_{0}\right]$. Next, observe that by Definition 1 for $q\leq q_{0}$ we have
$$\kappa_{2,q}\left(x,\epsilon \right)\;=\;\left\{\begin{array}{ccc} -\frac{x}{\phi_{\epsilon }\left(1-x\right)} & \mathrm{if} & -\frac{x}{\phi_{\epsilon }\left(1-x\right)}\geq \;q\\ \frac{\alpha_{0}-1}{\phi_{\epsilon }\left(\alpha_{0}\right)} & \mathrm{if} & -\frac{x}{\phi_{\epsilon }\left(1-x\right)}\leq q \end{array}\right.$$

This means that for $q<q^{*}$ we have $F\left(q\right)=\kappa_{2,q}\left(x,\epsilon \right)=-\frac{x}{\phi_{\epsilon }\left(1-x\right)}$, which is decreasing in *q*, and for for $q>q^{*}$ we have $F\left(q\right)=\frac{\alpha_{0}-1}{\phi_{\epsilon }\left(\alpha_{0}\right)}$, which increases in *q*. Finally, for $q=q^{*}$, we have $F\left(q\right)=-\frac{x}{\phi_{\epsilon }\left(1-x\right)}=q$.

It remains to verify that $\lim_{\epsilon \to 0}q\left(f,\epsilon \right)=2$. By the first claim of Lemma 9, $\phi_{0}\left(x\right)=\frac{x-1}{2}$. This means that for any 0<x≤1 we have $\lim_{\epsilon \to 0}\frac{-x}{\phi_{\epsilon }\left(1-x\right)}=2$. The claim follows since, by the preceding discussion, $q=q\left(f,\epsilon \right)=\frac{-x(q)}{\phi_{\epsilon }\left(1-x\left(q\right)\right)}$. □

## Data Availability

Not applicable.

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
