# Peer review of "Hypercontractive Inequalities for the Second Norm of Highly Concentrated Functions, and Mrs. Gerber’s-Type Inequalities for the Second Rényi Entropy"

_entropy, 2022, doi:10.3390/e24101376_

Round 1
Reviewer 1 Report
The paper establishes inequalities that upper bound the l2 norm and the entropy of a function of f_epsilon which is a noisy version of f defined on the hypercube {0,1}^n. The results are new and technically correct, although the techniques appear to have been developed before in [14]. The authors provide some examples where the new inequalities can be applied which was appreciated.
Author Response
It seems that no response is required
Reviewer 2 Report
The authors have targeted the submission to a rather narrow audience.
I am afraid that a large swath of the potential information theory audience will be turned off by the paper abstraction, which is unfortunate since the results are deep and potentially useful. For example, the authors use several times the phrase "an explicitly given function" without bothering to give the function. The authors should consider bringing the paper down to earth by explicitly writing those functions if they want to make the paper more useful to the research community.
Reviewer 3 Report
Dear Editor,
According to my knowledge and experience, the results presented in the paper are new and correct. So that I am glad to recommend the paper to your prestigious journal. Before acceptance of the paper, I suggest the following minor changing to make a better presentation of the paper:
1. Where is the motivation of the paper?! It's obligatory!
2. Are these new results sharp and more accurate compared with the others?
3. Please add the conclusion section.
4. By using a spell checker tool please check all the spelling of the paper.
5. I suggest adding some new relevant references to this topic.
Best Wishes
Author Response
The rseponse to this reviewer is contained in the responses to the editor and the other reviewers